# *CoFie*: Learning Compact Neural Surface Representations with Coordinate Fields

**Hanwen Jiang**   **Haitao Yang**   **Georgios Pavlakos**   **Qixing Huang**
Department of Computer Science, The University of Texas at Austin
{hwjiang,yanght,pavlakos,huangqx}@cs.utexas.edu

## Abstract

This paper introduces CoFie, a novel local geometry-aware neural surface representation. CoFie is motivated by the theoretical analysis of local SDFs with quadratic approximation. We find that local shapes are highly compressive in an aligned coordinate frame defined by the normal and tangent directions of local shapes. Accordingly, we introduce Coordinate Field, which is a composition of coordinate frames of all local shapes. The Coordinate Field is optimizable and is used to transform the local shapes from the world coordinate frame to the aligned shape coordinate frame. It largely reduces the complexity of local shapes and benefits the learning of MLP-based implicit representations. Moreover, we introduce quadratic layers into the MLP to enhance expressiveness concerning local shape geometry. CoFie is a generalizable surface representation. It is trained on a curated set of 3D shapes and works on novel shape instances during testing. When using the same amount of parameters with prior works, CoFie reduces the shape error by $48\%$ and $56\%$ on novel instances of both training and unseen shape categories. Moreover, CoFie demonstrates comparable performance to prior works when using even $70\%$ fewer parameters. Code and model can be found here: https://hwjiang1510.github.io/CoFie/

## 1   Introduction

In the realm of geometry modeling, neural implicit shape representations have become a powerful tool [33, 7, 4, 13, 41, 39, 2]. These representations typically use latent codes to represent shapes and employ multilayer perceptions (MLPs) to decode their Signed Distance Functions (SDFs). Early works in this field use a single latent code to represent an entire shape [33]. Nevertheless, the decoded SDFs usually lack geometry details. To improve the shape modeling quality, recent approaches have introduced local-based designs [4, 26, 42]. By decomposing an entire shape into many local surfaces, the shape modeling task becomes effortless – local surfaces are in simpler geometry which are easier to represent. Despite the progress, the local-aware design significantly increases the number of parameters, as each local surface is represented by one or even multiple latent codes. Thus, proposing a neural surface representation that is both *accurate and compact* is necessary.

To achieve this goal, we argue it is important to understand the properties of local surfaces. Following prior works [31, 49, 10, 44], we approximate the local geometry with quadratic patches [9] and perform analysis. Results show the feasibility of fitting the geometry of a specific category of quadratic patches. In detail, the quadratic patches are aligned with the coordinate system defined by the normal, principal directions, and principal curvatures of quadratic patch [9, 31]. However, when the quadratic patches are not aligned – they are freely transformed with random rotations and translations in 3D, mimicking real local surfaces – the optimization will be easily trapped into local

---

38th Conference on Neural Information Processing Systems (NeurIPS 2024).

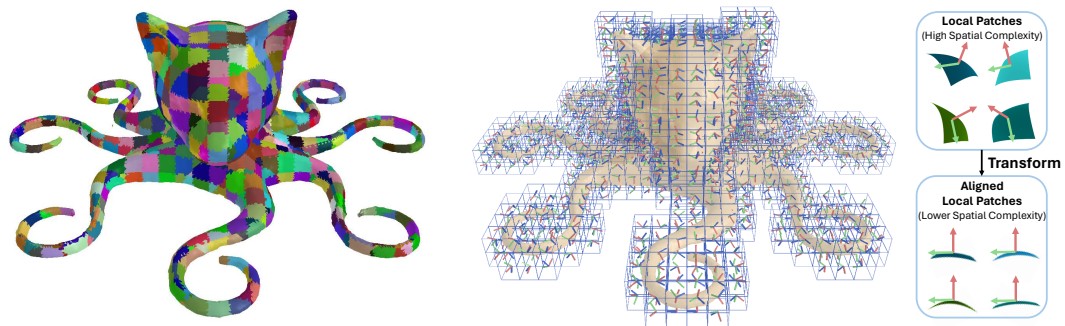

Figure 1: **CoFie is a local geometry-aware shape representation.** (Left) CoFie divides a shape into non-overlapping local patches, where each local patch is represented by an MLP-based Signed Distance Function. (Right) CoFie introduces Coordinate Field, which attaches a coordinate frame to each local patch. It transforms local patches from the world coordinate system to an aligned coordinate system, reducing shape complexity.

minima. This analysis reveals the difficulty of jointly recovering transformation information and geometry of local patches.

Based on the analysis, we propose CoFie, a novel local geometry-aware neural surface representation. The key insight of CoFie is **decomposing the transformation information of local shapes from its geometry**. As shown in Fig. 1, we associate each local surface with a learnable coordinate frame, which forms a Coordinate Field. We use the Coordinate Field to transform all local surfaces into an aligned coordinate system, reducing their spatial complexity. Thus, the geometry space of local surfaces becomes more compact, where the MLP-based neural SDFs are easier to learn.

An important design aspect is how to represent the Coordinate Field. Departing from the implicit-based representations, we use an explicit representation. Specifically, the coordinate frame of each local surface is parameterized by a rotation and a translation, forming a 6 Degree-of-Freedom pose. Moreover, we initialize the rotation using the estimated normal, principal direction, and principal curvature of a local surface. This design makes CoFie local geometry-aware and facilitates the learning of Coordinate Fields.

To better represent local surfaces' geometry, we introduce quadratic layers to the MLP. Prior works typically employ ReLU-based MLP with shallow layers and limited hidden size [33, 4]. Thus, the MLP is piece-wise linear [25] and cannot represent the distribution of local surfaces well. We demonstrate a simple quadratic layer improves the geometry modeling capability.

CoFie is a generalizable shape representation. After training on a curated dataset, it can represent arbitrary shapes that belong to any novel category. We evaluate CoFie on novel shape instances from both seen (training) and unseen categories, encompassing both synthetic and real shapes. Results show that CoFie outperforms prior arts, reducing the chamfer distance by $50\%$ on instances from both seen and unseen categories. Moreover, CoFie achieves comparable results with prior work using $70\%$ less parameters. In addition, we demonstrate that CoFie, which uses a single shared MLP for all shapes, achieves comparable results with methods that overfit a specific model for each testing shape.

## 2 Related Work

**Implicit Shape Representations.** Implicit shape representations are state-of-the-art in encoding shape geometric details [33, 41, 39, 3, 37, 53, 48, 52, 45, 28, 15, 55, 16, 34]. To improve the shape modeling capability, researchers inject local-aware designs. For example, DeepLS [4] integrates voxel grids and local MLPs to decode geometric shapes. Another line of work explores hierarchical representations where the local surfaces are divided unevenly [30, 47, 26, 42, 46, 40], leveraging Octree. For example, Multilevel Partition of Pnity [30] (MPU) blends parametric implicit surface patches into a global implicit surface. DOGNet uses dual-octree designs for neural MPU. The contribution of CoFie is perpendicular to these methods. CoFie still works on evenly divided voxels. However, instead of resolving high-frequency details of local shapes by using higher local resolution, CoFie proposes the Coordinate Field to reduce the spatial complexity. This is motivated by the analysis result that local geometric shapes are highly compressive under suitable coordinate frames.

The idea of a coordinate field is related to several existing approaches. For example, MVP [27] introduced oriented boxes for 3D face synthesis. However, in our setting, the variations in geometry and topology are much more significant than those of 3D human faces. Another relevant work is LDIF [18], which transforms a 3D point in the local coordinate system of each primitive to decode the iso-value of each shape. However, LDIF uses a fixed coordinate frame for each primitive. In contrast, the coordinate field varies spatially in CoFie and can be optimized, allowing us to capture detailed variations of the parts flexibly. Moreover, CoFie is based on a rigorous analysis of the expressivity of SDF and SDF learning. A follow-up approach [56] uses a warping field to transform a 3D point into a canonical space of specific categories. In contrast, CoFie is category-agnostic, benefiting from the use of local shapes.

On the learning side, many approaches show that MLPs are expressive and that their performance depends on the loss of training. For example, SAL [1] and SALD [2] show the importance of integrating normal losses to capture geometric features. SIREN [39] introduced other regularization losses to improve the quality of implicit representations learned. Although these approaches focus on local shape details, CoFie focuses on network design using coordinate frames. The CoFie approach is orthogonal to the encoding schemes.

**Hybrid 3D Representations.** Each 3D representation has fundamental advantages and limitations from the machine learning and representation perspective. For example, implicit representations allow flexible topologies, whereas explicit representations are easier to edit. Therefore, hybrid 3D representations, which aim to add the strength of different 3D representations for representation learning, have received a lot of attention. The main stream in hybrid 3D representations sequentially applies hybrid 3D models [24, 50, 12, 11, 38, 29, 8, 51]. For example, GRASS [24] combines a part-based representation to capture geometric structures of 3D shapes and a volumetric representation per part to capture geometric details of the parts. DSG-Net [50] employs part-based deformations to capture geometric details of the part. Other examples [12, 11, 38, 4, 14, 5, 19, 20] combine explicit graph, mesh, voxel and triplane representations with implicit volumetric representations to encode geometry details. CoFie is relevant to this series of approaches, where it combines voxel grids to encode global shapes and an implicit representation to decode local geometric details. The novelty of CoFie is that the local module employs a coordinate frame representation and enforces the prior knowledge that the local shape is roughly a low-complexity polynomial surface in the coordinate system defined by normal and principal directions.

**Coordinate Field Optimization.** The task of computing the proposed cell-based coordinate field is related to the problem of vector-field and frame-field design on meshes, where we want to ensure that the coordinate field is smooth and consistent among adjacent cells, and where we want the normal and tangent directions of each coordinate frame to align with the local fitting results if the fitting results are highly confident. This problem was studied in [35], which introduced a global optimization framework to compute a global vector field on a triangular mesh. Several more recent approaches have developed improved formulations for vector field optimization [17, 22] and extensions to frame field optimization [32, 36]. We refer to [43] for surveys on this topic. Rather than solving a global optimization problem to compute the coordinate field, the learning of the coordinate field in CoFie is driven by learning a compressive MLP.

## 3   Analysis of Fitting SDFs of Local Patches

In this section, we provide an analysis of fitting local surfaces. Following prior works [31, 49, 10, 44], we simplify local surfaces as quadratic patches. Additionally, we note that some works approximate local surfaces with linear patches [23, 47]. However, to handle the geometry details, it usually requires extremely high [47] or infinite resolution [23] during local surface partition. Approximating local surfaces with quadratic patch is more practical.

### 3.1   Importance of Non-linearity

A quadratic surface patch can be represented by $\boldsymbol{f}(u, v) = (u, v, \frac{1}{2}(au^2 + cv^2 + 2buv))$, where $u^2 + v^2 \leq r^2$ for locality, and $a, b, c$ are parameters for controlling the shape of the quadratic patch. The following proposition characterizes the SDF of a point $\boldsymbol{p}$ to $\boldsymbol{f}$.

**Proposition 1** *For each point $\boldsymbol{p} = (x, y, z)^T$ in the neighborhood of the origin $\boldsymbol{o}$, the signed distance function from $\boldsymbol{p}$ to $\boldsymbol{f}(u, v)$ can be approximated as*

$$d(\boldsymbol{p}, \boldsymbol{f}(u, v)) \approx z - \frac{1}{2}(ax^2 + cy^2 + 2bxy). \tag{1}$$

*where the approximation omits third-and-higher order terms in x, y, and z.*

*Proof:* See Appendix A.

Prop. 1 suggests that the SDF is non-linear. However, a shallow MLP using ReLU activation is piecewise linear, where the ReLU activation functions essentially decompose the input space into subspaces and the function in each subspace is still linear. This motivates the use of quadratic layers instead of linear layers (Sec. 4.2).

To hold generality, in Appendix B, we also analyze the local surface that can not be simplified as a single quadratic patch, i.e. sharp edges as the intersection of two quadratic patches.

### 3.2  Difficulty of Fitting Transformation Information

We demonstrate the difficulty of recovering the transformation information of quadratic patches during geometry fitting.

**Aligned Quadratic Patches.** Same as the previous section, we define the SDF of a quadratic local patch as $z - \frac{1}{2}(ax^2 + cy^2 + 2bxy)$, where the quadratic patch is axis-aligned. Consider a set of samples $\{((x_i, y_i, z_i), d_i), 1 \le i \le n\}$ from this quadratic patch, where $(x_i, y_i, z_i)$ is the location of the point $\boldsymbol{p}_i$, $d_i$ is the SDF value, and $n$ is the number of samples. To fit the surface from the samples, we solve the optimization problem as

$$\arg\min_{a,b,c} \sum_{i=1}^{n} \left( z_i - \frac{1}{2}(ax_i^2 + cy_i^2 + 2bx_iy_i) - d_i \right)^2 \tag{2}$$

which is a convex problem that has a unique global optimal.

**Unaligned Quadratic Patches.** Consider transforming the quadratic patch with a random rigid transformation $(R, \boldsymbol{t})$. This quadratic patch is not axis-aligned. In this case, the SDF function is given by $z' - \frac{1}{2}(ax'^2 + cy'^2 + 2bx'y')$ where $(x', y', z') = R(x, y, z) + \boldsymbol{t}$. To fit the surface from the samples, we solve the optimization problem as

$$\arg\min_{a,b,c,R,\boldsymbol{t}} \sum_{i=1}^{n} \left( z_i' - \frac{1}{2}(ax_i'^2 + cy_i'^2 + 2bx_i'y_i') - d_i \right)^2, \quad \begin{pmatrix} x_i' \\ y_i' \\ z_i' \end{pmatrix} = R \begin{pmatrix} x_i \\ y_i \\ z_i \end{pmatrix} + \boldsymbol{t}. \tag{3}$$

In this case, (3) becomes non-convex and has local minima. We defer a detailed characterization of the local minima of (3) to Appendix C.

In general, this non-convex problem makes geometry fitting non-trivial. It motivates the use of the Coordinate Field to explicitly model the transformation information and disentangle the transformation information of local patches from its geometry (Sec. 4.1).

## 4  CoFie

In this section, we introduce details of CoFie, including its representation (Sec. 4.1), MLP architecture (Sec. 4.2), and its learning scheme (Sec. 4.3).

### 4.1  CoFie Representation

As shown in Fig. 2, CoFie is based on a *hierarchical* representation, with coarse and fine-grained geometry. At the coarse level, it represents a shape with voxels. In detail, for a shape $\mathbf{S}$, it divides the space that contains the shape into $V \times V \times V$ non-overlapping voxel grids, where $V$ is the resolution of the voxel grids. A subset of voxels that intersect with the shape surface will be valid and CoFie only consider the valid sparse voxels to ensure its efficiency.

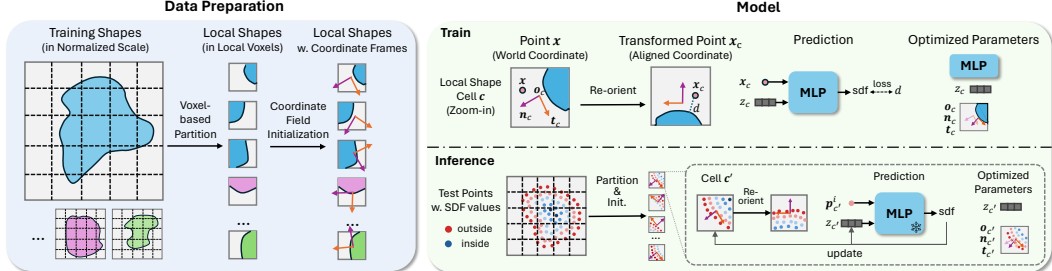

Figure 2: **Overview of CoFie**. CoFie represents a shape using a hybrid representation of voxels/cells and local implicit functions. (Left) For preparing the data for training the MLP-based local implicit functions, we split the training shapes into local shapes and initialize their coordinate frames using PCA. (Right) During training, a point will be transformed to the aligned coordinate of all local shapes using the coordinate frame. The MLP takes the transformed point and the latent code of the local shape to predict its SDF value. During testing, we fix the MLP, optimizing the latent codes and coordinate fields of valid cells.

At the fine-grained level, for each valid voxel $v$, we use an implicit representation to encode the geometry details for the local surface inside the voxel. Specifically, we use MLP-based neural SDFs. Each voxel $v$ has a latent code $z_v$ representing the local geometry and we use the MLP $g^\theta$ to decode the SDF values. For a point $x$, its SDF value contributed by the voxel $v$ is

$$f(\boldsymbol{x}, v) = g^\theta(\boldsymbol{x}_v, \boldsymbol{z}_v), \qquad \boldsymbol{x}_v = (\boldsymbol{n}_v, \boldsymbol{t}_v, \boldsymbol{n}_v \times \boldsymbol{t}_v)^T(\boldsymbol{x} - \boldsymbol{o}_v), \tag{4}$$

where $(\boldsymbol{o}_v, \boldsymbol{n}_v, \boldsymbol{t}_v)$ parameterize the coordinate frame of voxel $v$. Ideally, $\boldsymbol{o}_v$, $\boldsymbol{n}_v$ and $\boldsymbol{t}_v$ are the origin, normal direction, and tangent direction of the local surface, respectively.

Intuitively, for decoding the SDF value, we transform the point from the world coordinate system to the shared coordinate system for all local surfaces. $\boldsymbol{o}_v$ forms the translation between the two coordinate system, and $(\boldsymbol{n}_v, \boldsymbol{t}_v)$ form the rotation.

The final SDF value at $x$ is then given by

$$f(\boldsymbol{x}) = \frac{\sum_{v \in \mathcal{V}} w(\boldsymbol{x}, v) f(\boldsymbol{x}, v)}{\sum_{v \in \mathcal{V}} w(\boldsymbol{x}, v)}, \tag{5}$$

where $\mathcal{V}$ is the set of all valid voxels, $w(\boldsymbol{x}, v)$ is the weight assigned for the voxel $v$ with regard to point $x$. In practice, we use $w(\boldsymbol{x}, v) = 1$ if $\boldsymbol{x} \in v$, and $w(\boldsymbol{x}, v) = 0$ otherwise. Finally, the surface of a 3D shape is defined as the union set of local surfaces in its valid voxels $\mathcal{V}$.

## 4.2 CoFie MLP Architecture

Following the common practice of MLP, we define

$$g^\theta(\boldsymbol{x}, \boldsymbol{z}) = g_L^{\theta_L} \circ \phi \circ \boldsymbol{g}_{L-1}^{\theta_{L-1}} \circ \phi \cdots \circ \phi \circ \boldsymbol{g}_1^{\theta_1}(\boldsymbol{x}, \boldsymbol{z})$$

where $\boldsymbol{g}_l^{\theta_l} : \mathbb{R}^{m_{l-1}} \to \mathbb{R}^{m_l}$ is a layer with trainable parameters $\theta_l$, and where $\phi$ is an activation function. Denote $\boldsymbol{z}_l$ as the output in layer $l$, i.e., $\boldsymbol{z}_0 = (\boldsymbol{x}; \boldsymbol{z})$. A common strategy is to set each $\boldsymbol{g}_l^{\theta_l}$ as a linear function, i.e.,

$$\boldsymbol{g}_l^{\theta_l}(\boldsymbol{z}_{l-1}) = A_l \boldsymbol{z}_{l-1} + \boldsymbol{b}_l, \tag{6}$$

where $\theta_l = (A_l, \boldsymbol{b}_l)$. Furthermore, $\phi$ is chosen as the ReLU layer, i.e., $\phi(\boldsymbol{z}_l) = \max(\boldsymbol{z}_l, \boldsymbol{0})$ where the max operator is applied element-wise. This strategy is widely used in prior works [33, 7].

However, in Sec. 3.1, we demonstrate the SDF function has non-negligible quadratic components locally and its incompatibility with MLPs with linear layers and ReLU activation. Therefore, instead, we model the quadratic components with quadratic layers. We let the top $k$ layers of $g^\theta$ to be quadratic functions, where $k \geq 1$. The quadratic layer can be formulated as

$$\boldsymbol{g}_l^{\theta_l}(\boldsymbol{z}_{l-1}) = \boldsymbol{z}_{l-1}^T T_l \boldsymbol{z}_{l-1} + A_l \boldsymbol{z}_{l-1} + \boldsymbol{b}_l \tag{7}$$

where $T_l \in \mathbb{R}^{m_{l-1} \times m_l \times m_{l-1}}$ is a tensor, and $\theta_l = (T_l, A_l, \boldsymbol{b}_l)$.

We can understand the trade-offs between the use of linear layers (Eq. 6) and the quadratic layers (Eq. 7) as follows. With the same latent dimensions $m_l$, the quadratic layers have many more parameters than the linear layers. Therefore, with the same network size, we have to use fewer layers or smaller latent dimensions for quadratic layers. This will limit the capability of the network instead. In practice, setting $k = 1$ leads to the best performance.

### 4.3  CoFie Learning Scheme

**Problem Setup.** Following DeepSDF-series, we perform **shape auto-decoding** [33, 4]. The task assesses the capability of models to fit/represent given shapes. During both training and inference, the input is points sampled freely in space with their ground-truth SDF values. The output is the neural SDF. Additionally, we notify that the task is different from shape reconstruction from point cloud inputs, or so-called **shape auto-encoding**, which is studied in [54, 8, 29].

Moreover, CoFie is a **generalizable** shape representation. It is trained on a curated dataset with multiple shapes. Once trained, the MLP can be used to represent or decode the SDF of any incoming shapes. We note the setting of generalizable shape representation is **different from overfitting a shape**, where an MLP is specialized for each shape.

**Training and Inference.** We follow the protocol of the shape auto-decoding task [33, 4]. We train CoFie with a set of shapes denoted as $\mathcal{S} = \{S_i, 1 \le i \le n\}$. For each shape, we perform voxelization (Sec. 4.1) and train CoFie with valid local shapes. We denote the set of valid local shapes of shape $S_i$ as $\mathcal{V}_i$. Following [33, 39, 4], we collect a set of point samples $\mathcal{P}_v = (\boldsymbol{p}^j, d^j)$ in the neighborhood of each voxel $v \in \mathcal{V}_i$, where $\boldsymbol{p}^j$ and $d^j$ denote the position of the sample and the SDF value of $\boldsymbol{p}^j$. The point samples are sampled in free space and are not necessary to be on-surface points. For each local shape in voxel $v$, we associate it with a latent code $\boldsymbol{z}_v$ and the coordinate frame $(\boldsymbol{o}_v, \boldsymbol{n}_v, \boldsymbol{t}_v)$. Then the training objective can be formulated as

$$\underset{\theta, \{\boldsymbol{o}_v, \boldsymbol{n}_v, \boldsymbol{t}_v, \boldsymbol{z}_v | v \in \mathcal{V}_i\}}{\arg\min} \sum_{i=1}^{n} \sum_{v \in \mathcal{V}_i} \sum_{(\boldsymbol{p}^j, d^j) \in \mathcal{P}_v} ||g^\theta(\boldsymbol{p}_v^j, \boldsymbol{z}_v) - d^j||_1, \tag{8}$$

where $\boldsymbol{p}_v^j = (\boldsymbol{n}_v, \boldsymbol{t}_v, \boldsymbol{n}_v \times \boldsymbol{t}_v)^T (\boldsymbol{p}^j - \boldsymbol{o}_v)$. In this step, we jointly optimize the MLP, the latent codes, and the coordinate field for all training shapes. Intuitively, it trains the MLP to represent training shapes and optimize the compatibility between the MLP, latent codes and the coordinate fields.

During inference, we freeze the MLP $g^\theta$. We optimize the latent code and the coordinate field for a single target shape at one time. It is formulated as

$$\underset{\{\boldsymbol{z}_v, \boldsymbol{o}_v, \boldsymbol{n}_v, \boldsymbol{t}_v | v \in \mathcal{V}\}}{\arg\min} \sum_{v \in \mathcal{V}} \sum_{(\boldsymbol{p}^j, \boldsymbol{n}^j) \in \mathcal{P}_v} ||g^\theta(\boldsymbol{p}_v^j, \boldsymbol{z}_v) - d^j||_1 \tag{9}$$

Besides, we use the regularization term over the inferred latent codes following [33, 4].

**Shape Consistency at Boundary of Voxels.** If we sample the points $\mathcal{P}_v$ within each voxel $v$, Eq. 8 and Eq. 9 optimize the local geometry within each voxel independently. This may lead the non-smooth and inconsistency surface at the boundary of voxels. To solve this, we follow [4] to expand receptive field of each voxel by sampling points from their neighbouring voxels.

**Coordinate Field Initialization.** Eq. 8 has many unwanted local minima, especially for optimizing the coordinate field. Thus, a good initialization of the coordinate fields ensures the compactness of local shape at early stage of training, and facilitates the learning of MLP. Motivated by the analysis in Sec. 3.2, we use estimated normal and tangent directions to initialize the coordinate fields. In detail, we compute the derivatives of SDF values at these point samples and perform PCA to get them. Besides, $\boldsymbol{o}_c$ is initialized as the center of the cell. We find that this initialization is important to reduce errors (Sec. 5.2).

## 5  Experiment

This section presents an experimental evaluation of CoFie. We begin with the experimental setup and then present the results and ablations.

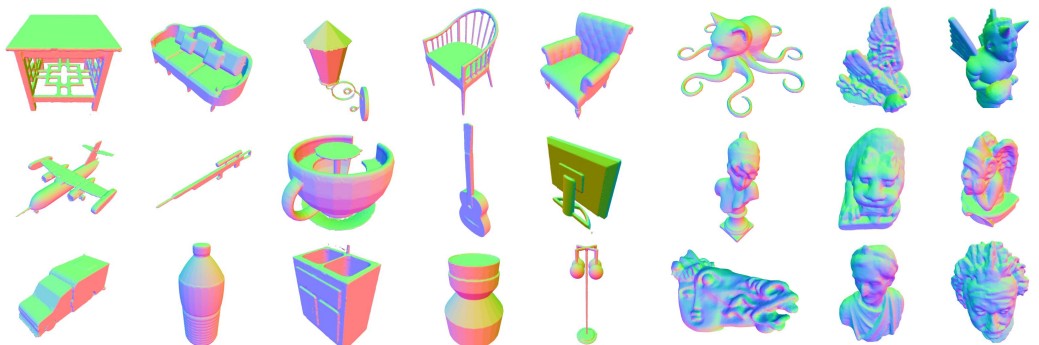

Figure 3: Diveristy and quality of meshes that CoFie can represent. The results include both novel instances from ShapeNet training categories (top left), instances from ShapeNet unseen categories (bottom left), and real shapes from the Thingi dataset (right). We visualize the shapes with surface normal to better show their geometry. Please see the appendix for comparisons with ground-truth.

**Implementation Deatils.** We use latent code of size 125 for all cells. The MLP is composed of 5 layers where the first 4 layers are linear layers and the last layer is quadratic. The hidden channel size is 128. We use the voxel grid size of $32 \times 32 \times 32$. During training, we use 12 shapes for each batch. For each shape, we sample 3000 voxels that intersect with the surface of the shape (with return). We sample 24 points for each cell for training, and each point is sampled within 1.5 times the radius of the voxel to ensure boundary consistency between cells. We use the Adam optimizer [21] with learning rates $5e - 4$, $1e - 3$, and $1e - 3$ for the MLP, coordinate fields, and latent codes. We train with 150000 iterations and reduce the learning rates by half for every 20000 iteration. During inference, we use a learning rate of $5e - 4$ for 800 iterations. Reconstructed meshes are obtained by performing Marching Cubes with a 128 resolution by default. We use the quaternion representation for the rotation matrix of the coordinate frames. We train on 4 GPUs with 24GB memory for 1 day.

**Training and testing data.** We train CoFie on 1000 shape instances sample from ShapeNet [6] of chairs, planes, tables, lamps, and sofas (200 instances for each category). We test CoFie with three test sets for comprehensive analysis of CoFie: i) 250 novel instances from the 5 training ShapeNet categories; ii) 250 novel instances from 10 unseen ShapeNet categories; iii) 24 meshes from the Thingi dataset [57], which captures real scenes. The test set i) checks how CoFie fits the training distribution. Test sets ii) and iii) are used to test the generalization capability of CoFie on novel shapes that observe different structures with training shapes.

**Baseline Approaches** We compare our CoFie with three types of methods: *generalizable methods*, which use a single MLP to represent multiple shapes; *shape-specific methods*, which train an MLP for each testing shape. Generally, the latter genre demonstrates a better performance as the MLP model can be trained to overfit a single testing shape. Both the two types of methods performs shape auto-decoding. Besides, we also report results for a state-of-the-art shape auto-encoding method. We note that it is a reference method while the result is not directly comparable.

Note that CoFie is a generalizable method for shape auto-decoding. We include more details for baselines as follows.

- **DeepSDF** [33] is a generalizable shape auto-decoding method using a global latent code to represent one shape.

- **DeepLS** [4] is a generalizable shape auto-decoding method using local-based representations. DeepLS is a direct comparable baseline.

- **NGLOD** [40] is a shape-specific method for shape auto-decoding, achieving state-of-the-art performance. For a fair comparison with CoFie, we use the level of detail as 3, keeping the number of parameters of the latent codes in the same magnitude as our CoFie.

- **3DS2VS** [54] is a generalizable shape auto-encoding method. It employs transformers to predict the shape latent code, rather than getting it by optimization (shape auto-decoding). The input is on-surface point clouds.

Table 1: Shape errors on novel instances of the ShapeNet training categories. We report chamfer distance ($10^{-4}$) and highlight the best.

| | Novel Instances of Seen Shape Category | | | | | |
|---|---|---|---|---|---|---|
| | chair | lamp | plane | sofa | table | mean |
| 3DS2VS | 9.11 | 10.9 | 1.68 | 8.76 | 13.7 | 8.85 |
| DeepSDF | 5.69 | 15.1 | 7.51 | 4.08 | 6.64 | 7.84 |
| DeepLS | 7.70 | 6.57 | 0.83 | 2.54 | 2.18 | 3.91 |
| CoFie (ours) | **2.35** | **3.13** | **0.80** | **2.44** | **1.41** | **2.05** |

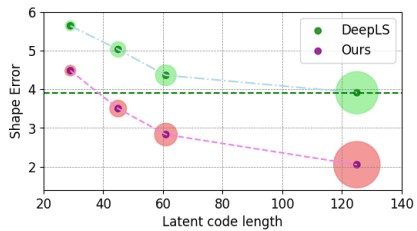

Figure 4: Trade-off between accuracy and model size ( notified by the radius of circles).

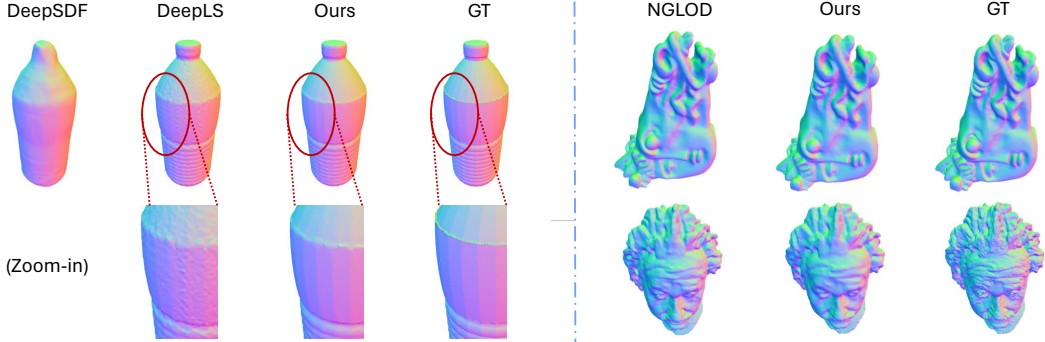

Figure 5: Comparison with prior works. (Left) Results of generalizable methods, where our CoFie demonstrates better capability for modeling geometry details. (Right) Compare with the per-shape-based method NGLOD. We note that NGLOD is a shape-specific method that overfits one MLP on one testing shape.

Besides, we also compare with state-of-the-art shape auto-encoding (point cloud reconstruction) methods. We note these methods are used as reference for understanding the model performance. They are not directly comparable.

We train DeepSDF, DeepLS, and CoFie using the same dataset for fair comparisons. NGLOD is trained on each test shape. All methods receive the same inputs during inference.

**Evaluation Metrics** We report the mesh reconstruction error as the chamfer$-L_2$ distance between the reconstructed and ground-truth meshes. We sample 30000 points to compute the chamfer distances. The meshes are normalized into a unit scale.

## 5.1 Experimental Results

**Qualitative Results.** As shown in Fig. 3, CoFie demonstrates strong surface representation capability. The details of geometry are maintained well. The results on out-of-distribution shapes from unseen categories are comparable to the training categories.

**Performance on Training Categories.** As shown in Table 1, CoFie outperforms baselines by a large margin. In detail, the average chamfer distance of CoFie is 1.86 (48% relatively) smaller than the best baseline DeepLS. Moreover, we provide a more detailed comparison with DeepLS, as shown in Fig. 4. We observe that CoFie is consistently better than DeepLS with different latent code and MLP size. Specifically, CoFie with latent code size 48 achieves slightly better performance compared with DeepLS with latent code size 128. Note that the number of MLP parameters for the former is about 15% for the latter.

Table 2: Shape errors on instances of the ShapeNet novel categories. We evaluate the chamfer distance ($10^{-4}$).

| | Unseen Shape Category | | | | | | | | | | |
|---|---|---|---|---|---|---|---|---|---|---|---|
| | cabinet | car | phone | bus | guitar | clock | bottle | mug | washer | rifle | mean |
| 3DS2VS | 16.4 | 12.7 | 21.9 | 24.1 | 2.4 | 10.5 | 10.6 | 9.3 | 26.7 | 25.3 | 16.0 |
| DeepSDF | 12.3 | 6.87 | 6.92 | 18.4 | 11.8 | 10.6 | 4.54 | 10.83 | 6.17 | 15.7 | 10.4 |
| DeepLS | 9.74 | 5.77 | 2.09 | 7.22 | **0.63** | 4.30 | 12.7 | 7.28 | 18.8 | 4.13 | 7.27 |
| CoFie (ours) | **4.19** | **3.09** | **1.86** | **3.66** | 1.23 | **3.57** | **4.48** | **2.58** | **4.23** | **2.88** | **3.18** |

Table 3: Results on Thingi meshes. We evaluate the chamfer distance ($10^{-4}$) with a marching cube resolution of 256. Note that *NGLOD is trained on each test shape*, while CoFie uses a shared MLP for all shapes as a generalizable method.

| | Unseen Thingi Shapes | | |
|---|---|---|---|
| | Generalizable | Total MLP Size | Shape Error |
| NGLOD | ✗ | 24 ×0.2MB | **1.04** |
| DeepSDF | ✓ | 0.2MB | 3.68 |
| CoFie (ours) | ✓ | 0.2MB | 1.87 |

Table 4: Ablation study of (0) Base performance; (1) coordinate field and its initialization methods; (2) using quadratic MLP; (3) full performance. We use resolution 128 to get reconstructed meshes.

| | Coord. Field (CF) | | MLP Settings | | Error |
|---|---|---|---|---|---|
| | Use CF | Geo-Aware Init. | # Linear | # Quad. | |
| (0) | ✗ | ✗ | 5 | ✗ | 3.91 |
| (1) | ✓ | ✗ | 5 | ✗ | 3.45 |
| | ✓ | ✓ | 5 | ✗ | 2.33 |
| (2) | ✗ | ✗ | 5 | 1 | 3.01 |
| | ✗ | ✗ | 6 | ✗ | 3.70 |
| (3) | ✓ | ✓ | 5 | 1 | **2.05** |

**Performance on Unseen Categories.** We compare CoFie with previous generalizable methods on ShapeNet unseen categories and the state-of-the-art per-shape-based method on the challenging real scans. We provide visualization results in Fig. 5.

- *ShapeNet Unseen Categories.* As shown in Table 2, CoFie achieves better generalization on 9 out of 10 novel shape categories. We also observe that the performance gap between CoFie and prior works is larger in the unseen categories, showing the strong generalization capability of CoFie.
- *Thingi Real Shapes.* As shown in Table 3, CoFie achieves comparable results with NGLOD. We note that NGLOD is a per-shape-based method, which trains a model for each shape and performs better naturally. In contrast, CoFie is trained on ShapeNet shapes.

## 5.2 Ablation Study

As shown in Table 4, we experiment with CoFie variants to validate the effectiveness of our coordinate field and MLP designs.

**Coordinate Field and Initialization.** As shown in Table 4 (1), using coordinate fields with different initialization strategies can both reduce the shape error. In detail, when using axis-aligned coordinate field initialization, where all coordinate frames are initialized as the world frame, the shape error reduced slightly from 3.91 to 3.45. The result demonstrates the difficulty of optimizing coordinate frames. In contrast, when using geometry-aware initialization, i.e., initializing local frames with estimated normal and tangent directions of local shapes, the shape error is reduced to 2.33, observing a 40% improvement.

**MLP Design.** As shown in Table 4 (2), using a quadratic layer as the last layer of the MLP observes a 0.9 (23% relatively) reduction of shape error. As the use of the quadratic layer introduces additional parameters, we compare it with a variant for a fair comparison. In detail, we compare it with a linear MLP with an additional layer (6 layers in total), where the two MLPs have the same amount of parameters because the output channel size of the last layer is 1. The result shows that increasing the number of linear layers can only reduce the shape error slightly.

Moreover, Table 4 (3) demonstrates the combination of the two introduced techniques can jointly reduce the shape error.

## 6 Conclusions and Future Work

This paper has introduced CoFie, a novel neural surface representation. It is based on the theoretical results of using a ReLU-based MLP to encode geometric shapes. The results strongly motivate

the use of local coordinate frames, which encompass the coordinate fields, to transform a point before decoding its SDF value using an MLP. This leads to a hybrid representation combined with coordinate frames associated with local voxels. The experimental results show a strong generalization behavior of CoFie in new instances for shape reconstruction, which significantly outperforms previous generalizable methods and achieves comparable results to shape-specific methods.

**Limitations.** One limitation of CoFie is that it is based on local shapes and cannot be used for the shape completion task. Different from DeepSDF, which learns global shape priors and can fill the large missing components in the input, CoFie is restricted to observable parts. We plan to incorporate more global priors into CoFie. Besides, with a fixed cell resolution, the local shape analysis is broken when a local cell intersects with thin structures. We plan to extend it with adaptive local cell resolutions.

**Broader Impact.** CoFie is a neural surface representation, which have the potential to be used for 3D reconstruction and generation.

**Acknowledgment.** Q. H. would like to acknowledge NSF IIS 2047677 and NSF IIS 2413161

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

# Appendices

## A  Proof of Prop. 1

Let $(u, v)$ be the parameters of the closest point of $\boldsymbol{p} = (x, y, z)^T$ on $\boldsymbol{f}(u, v)$. We have the following constraints on $(u, v)$:

$$(\boldsymbol{p} - \boldsymbol{f}(u, v))^T \boldsymbol{f}_u(u, v) = 0 \tag{10}$$

$$(\boldsymbol{p} - \boldsymbol{f}(u, v))^T \boldsymbol{f}_v(u, v) = 0 \tag{11}$$

Note that

$$\boldsymbol{f}_u(u, v) = (1, 0, au + bv)^T$$
$$\boldsymbol{f}_v(u, v) = (0, 1, bu + cv)^T$$

Ignoring quadratic-and-higher order terms in $u, v, x, y$, and $z$ in (10) and (11), we have

$$(x - u) + z(au + bv) \approx 0 \tag{12}$$

$$(y - v) + z(bu + cv) \approx 0 \tag{13}$$

This leads to

$$\begin{pmatrix} u \\ v \end{pmatrix} = \begin{pmatrix} 1 - az & -bz \\ -bz & 1 - cz \end{pmatrix}^{-1} \begin{pmatrix} x \\ y \end{pmatrix}$$

$$\approx \begin{pmatrix} x \\ y \end{pmatrix} + \begin{pmatrix} a & b \\ b & c \end{pmatrix} \begin{pmatrix} x \\ y \end{pmatrix} z. \tag{14}$$

The normal direction at $(u, v)$ is

$$\begin{aligned}
\boldsymbol{n}(u, v) &= \frac{\boldsymbol{f}_u(u, v) \times \boldsymbol{f}_v(u, v)}{\|\boldsymbol{f}_u(u, v) \times \boldsymbol{f}_v(u, v)\|} \\
&= \frac{(-(au + bv), -(bu + cv), 1)^T}{\sqrt{1 + (au + bv)^2 + (bu + cv)^2}}.
\end{aligned} \tag{15}$$

The signed-distance function of $\boldsymbol{p}$ to $\boldsymbol{f}(u, v)$ is given by

$$d(\boldsymbol{p}, \boldsymbol{f}(u, v)) = (\boldsymbol{p} - \boldsymbol{f}(u, v))^T \boldsymbol{n}(u, v). \tag{16}$$

Substituting (15), (14) into (16) and ignoring third-and-higher terms in $u, v, x, y, z$, we have

$$\begin{aligned}
d(\boldsymbol{p}, \boldsymbol{f}(u, v)) &\approx -(x - u)(au + bv) - (y - v)(bu + cv) \\
&\quad + \left(z - \frac{1}{2}(au^2 + 2buv + cv^2)\right) \\
&\approx z - \frac{1}{2}(au^2 + 2buv + cv^2)
\end{aligned}$$

$\square$

## B  Representing Sharp Edges as Quadratic Patches

We consider the intersection of two quadratic patches where the intersection is along the $y$-axis. In this case, we can define the surface patch as $\boldsymbol{f}(u, v) = (u, v, f(u, v))^T$ where

$$f(u, v) = \begin{cases} \frac{1}{2}(a_1 u^2 + c_1 v^2 + 2b_1 uv) + e_1 u & u \leq 0 \\ \frac{1}{2}(a_2 u^2 + c_1 v^2 + 2b_2 uv) + e_2 u & \text{otherwise} \end{cases} \tag{17}$$

In (17), we do not have any linear term in $v$, so that the normals to these two patches at $(0, 0, 0)^T$ are in the $xz$ plane. In addition, the coefficients in front of $v^2$ are identical, so these two patches stitch along $u = 0$.

The following proposition provides an approximation to the SDF function of $\boldsymbol{f}(u, v)$.

**Proposition 2** *For each point $\boldsymbol{p} = (x, y, z)^T$ in the neighborhood of the origin $\boldsymbol{o}$, the signed distance function from $\boldsymbol{p}$ to $\boldsymbol{f}(u, v)$ can be approximated as*

$$d(\boldsymbol{p}, \boldsymbol{f}(u,v)) \approx \begin{cases} z - \frac{1}{2}(a_1 x^2 + c_1 y^2 + 2b_1 xy) - e_1 x & x \le 0 \\ z - \frac{1}{2}(a_2 x^2 + c_1 y^2 + 2b_2 xy) - e_2 x & \text{otherwise} \end{cases}. \tag{18}$$

The proof is very similar to that of Prop. 1. When $x \ge 0$, the parameters $(u, v)$ of the closest point satisfy $u \ge 0$, and vice versa. Therefore, the proof applies the description in Section A. □

## C   Local Minima of (3)

We will show that there are nontrivial local minima due to symmetries induced by the rotation group. However, those local minima do not recover the underlying ground-truth shape. As a result, they force the network to learn the wrong patterns from the data. For simplicity, we focus on the 2D setting. The extension to 3D is straightforward.

In 2D, we assume that the underlying curve is $(x, k_0 x^2)$. SDF samples are given by $(x, k_0 x^2 + y, y)$ where $x \sim p, y \sim q$. Consider the 2D rigid pose parameters $\theta, t_x, t_y$. Let $k$ be the curve parameter. Our goal is to optimize parameters $\theta, t_x, t_y, k$ to minimize the following $L^2$ reconstruction loss:

$$r(k, t_x, t_y, \theta) = \mathop{\mathbb{E}}_{x \sim p} \mathop{\mathbb{E}}_{y \sim q} \Big( \sin(\theta)x + \cos(\theta)(k_0 x^2 + y) + t_y - $$
$$k\big( \cos(\theta)x - \sin(\theta)(k_0 x^2 + y) + t_x \big)^2 - y \Big)^2$$

Clearly, $(k_0, 0, 0, 0)$ is a global minimum of $r$. The following proposition shows that there is another local minimum of $r$.

**Proposition 3** *Suppose $p$ and $q$ are independent, and*

$$\mathop{\mathbb{E}}_{x \sim p} x = 0.$$

*Then $(-k_0, 0, 2c, \pi)$ is a critical point of $r$, where $c = \mathop{E}_{y \sim q} y$. In addition, it is a local minimum of $r$ if we assume*

$$\mathop{\mathbb{E}}_{x \sim p} x^3 = \mathop{\mathbb{E}}_{x \sim p} x^5 = 0, \quad |y| \ll |x|.$$

We defer the proof of Prop. 3 to Appendix C.1. Prop. 3 shows that there is a non-trivial critical point whose parameters depend on the sampling pattern. As neural network training mostly uses first-order methods that can be trapped into critical points, this means that without careful initialization, the network will memorize non-shape-related patterns from data, and significantly impairs the generalization ability of the resulting network.

### C.1   Proof of Prop. 3

Denote

$$l(x, y, k, t_x, t_y, \theta) = \sin(\theta)x + \cos(\theta)(k_0 x^2 + y) + t_y - $$
$$k\big( \cos(\theta)x - \sin(\theta)(k_0 x^2 + y) + t_x \big)^2 - y.$$

It is easy to check that

$$l(x, y, -k_0, 0, 2c, \pi) = 2c - 2y. \tag{19}$$

The first-order gradients of $l$ with respect to $k, t_x, t_y, \theta$ are given by

$$\frac{\partial l}{\partial k}(x, y, -k_0, 0, 2c, \pi) = -x^2, \tag{20}$$

$$\frac{\partial l}{\partial t_x}(x, y, -k_0, 0, 2c, \pi) = -k_0 x, \tag{21}$$

$$\frac{\partial l}{\partial t_y}(x, y, -k_0, 0, 2c, \pi) = 1, \tag{22}$$

$$\frac{\partial l}{\partial \theta}(x, y, -k_0, 0, 2c, \pi) = -x - k_0^2 x^3 - k_0 x y. \tag{23}$$

Therefore, we have

$$\frac{\partial r}{\partial k}(-k_0, 0, 2c, \pi) = \mathbb{E}_{x \sim p} \mathbb{E}_{y \sim q} \frac{\partial l}{\partial k}(x, y, -k_0, 0, 2c, \pi) l(x, y, -k_0, 0, 2c, \pi)$$
$$= -\mathbb{E}_{x \sim p} \mathbb{E}_{y \sim q} (2c - 2y) x^2 = 0,$$

and

$$\frac{\partial r}{\partial t_x}(-k_0, 0, 2c, \pi) = \mathbb{E}_{x \sim p} \mathbb{E}_{y \sim q} \frac{\partial l}{\partial t_x}(x, y, -k_0, 0, 2c, \pi) l(x, y, -k_0, 0, 2c, \pi)$$
$$= \mathbb{E}_{x \sim p} \mathbb{E}_{y \sim q} (2c - 2y) kx = 0,$$

and

$$\frac{\partial r}{\partial t_y}(-k_0, 0, 2c, \pi) = \mathbb{E}_{x \sim p} \mathbb{E}_{y \sim q} \frac{\partial l}{\partial t_y}(x, y, -k_0, 0, 2c, \pi) l(x, y, -k_0, 0, 2c, \pi)$$
$$= \mathbb{E}_{x \sim p} \mathbb{E}_{y \sim q} (2c - 2y) = 0,$$

and

$$\frac{\partial r}{\partial t_x}(-k_0, 0, 2c, \pi) = \mathbb{E}_{x \sim p} \mathbb{E}_{y \sim q} \frac{\partial l}{\partial t_x}(x, y, -k_0, 0, 2c, \pi) l(x, y, -k_0, 0, 2c, \pi)$$
$$= -\mathbb{E}_{x \sim p} \mathbb{E}_{y \sim q} (2c - 2y)(x + k_0^2 x^3 + k_0 x y) = 0.$$

This means that $(-k_0, 0, 2c, \pi)$ is a critical point of $r$. To show that it is indeed a local minimum, we study the second-order derivatives of $r$. We begin with the second-order derivatives of $l$. They are

$$\frac{\partial^2 l}{\partial^2 \theta}(x, y, -k_0, 0, 2c, \pi) = 2k_0(k_0 x^2 + y)^2 + y - k_0 x^2$$

$$\frac{\partial^2 l}{\partial \theta \partial t_x}(x, y, -k_0, 0, 2c, \pi) = 2k_0(k_0 x^2 + y)$$

$$\frac{\partial^2 l}{\partial^2 t_x}(x, y, -k_0, 0, 2c, \pi) = 2k_0,$$

and

$$\frac{\partial^2 l}{\partial^2 k}(x, y, -k_0, 0, 2c, \pi) = 0$$

$$\frac{\partial^2 l}{\partial k \partial t_x}(x, y, -k_0, 0, 2c, \pi) = 2x$$

$$\frac{\partial^2 l}{\partial k \partial \theta}(x, y, -k_0, 0, 2c, \pi) = 2x(k_0 x^2 + y),$$

and

$$\frac{\partial^2 l}{\partial^2 t_y}(x, y, -k_0, 0, 2c, \pi) = 0, \qquad \frac{\partial^2 l}{\partial t_y \partial k}(x, y, -k_0, 0, 2c, \pi) = 0$$

$$\frac{\partial^2 l}{\partial t_y \partial t_x}(x, y, -k_0, 0, 2c, \pi) = 0, \qquad \frac{\partial^2 l}{\partial t_y \partial \theta}(x, y, -k_0, 0, 2c, \pi) = 0$$

Note that $\forall \alpha, \beta \in \{k, t_x, t_y, \theta\}$,

$$\frac{\partial^2 r}{\partial \alpha \partial \beta}(-k_0, 0, 2c, \pi)$$

$$= \mathop{\mathbb{E}}_{x \sim p} \mathop{\mathbb{E}}_{y \sim q} \left( l(x, y, -k_0, 0, 2c, \pi) \frac{\partial l^2}{\partial \alpha \partial \beta}(x, y, -k_0, 0, 2c, \pi) \right.$$

$$\left. + \frac{\partial l}{\partial \alpha}(x, y, -k_0, 0, 2c, \pi) \frac{\partial l}{\partial \beta}(x, y, -k_0, 0, 2c, \pi) \right)$$

Denote

$$V_x^i = \mathop{\mathbb{E}}_{x \sim p} x^i, \qquad V_y^i = \mathop{\mathbb{E}}_{y \sim q} y^i.$$

We have

$$\frac{\partial^2 r}{\partial^2 \theta}(-k_0, 0, 2c, \pi) = V_x^2 + 2k_0 c(V_x^2 + 2V_y^2) + (2 + 4V_x^2 k_0^2)(c^2 - V_y^2)$$

$$- 4k_0 V_y^3 + k_0^2(2V_x^4 + V_x^2 V_y^2) + 2k_0^3 c V_x^4 + k_0^4 V_x^6$$

$$\frac{\partial^2 r}{\partial \theta \partial t_x}(-k_0, 0, 2c, \pi) = k_0 \left( V_x^2 + k_0^2 V_x^4 + k_0 c V_x^2 + 4(c^2 - V_y^2) \right)$$

$$\frac{\partial^2 r}{\partial^2 t_x}(-k_0, 0, 2c, \pi) = k_0^2 V_x^2,$$

and

$$\frac{\partial^2 r}{\partial^2 k}(-k_0, 0, 2c, \pi) = V_x^4$$

$$\frac{\partial^2 r}{\partial k \partial t_x}(-k_0, 0, 2c, \pi) = k_0 V_x^3 = 0$$

$$\frac{\partial^2 r}{\partial k \partial \theta}(-k_0, 0, 2c, \pi) = V_x^3(1 + k_0 c) + k_0^2 V_x^5 = 0,$$

and

$$\frac{\partial^2 r}{\partial^2 t_y}(-k_0, 0, 2c, \pi) = 1,$$

$$\frac{\partial^2 r}{\partial t_y \partial k}(-k_0, 0, 2c, \pi) = -V_x^2$$

$$\frac{\partial^2 r}{\partial t_y \partial t_x}(-k_0, 0, 2c, \pi) = 0,$$

$$\frac{\partial^2 r}{\partial t_y \partial \theta}(-k_0, 0, 2c, \pi) = 0$$

It remains to show that

$$\frac{\partial^2 r}{\partial^2 \theta}(-k_0, 0, 2c, \pi) \frac{\partial^2 r}{\partial^2 t_x}(-k_0, 0, 2c, \pi) > \left( \frac{\partial^2 r}{\partial \theta \partial t_x}(-k_0, 0, 2c, \pi) \right)^2 \tag{24}$$

and

$$\frac{\partial^2 r}{\partial^2 k}(-k_0, 0, 2c, \pi) \frac{\partial^2 r}{\partial^2 t_y}(-k_0, 0, 2c, \pi) > \left( \frac{\partial^2 r}{\partial k \partial t_y}(-k_0, 0, 2c, \pi) \right)^2 \tag{25}$$

The difference between the left and right-hand sides of (24)

$$k_0^2 \Big( V_x^2 \big( V_x^2 + 2k_0 c(V_x^2 + 2V_y^2) + (2 + 4V_x^2 k_0^2)(c^2 - V_y^2) - 4k_0 V_y^3 $$
$$+ k_0^2(2V_x^4 + V_x^2 V_y^2) + 2k_0^3 c V_x^4 + k_0^4 V_x^6 \big) - \big( V_x^2 + k_0^2 V_x^4 \big)^2 $$
$$- \big( k_0 c V_x^2 + 4(c^2 - V_y^2) \big)^2 - 2 \big( V_x^2 + k_0^2 V_x^4 \big) \big( k_0 c V_x^2 + 4(c^2 - V_y^2) \big) \Big) $$
$$= k_0^2 \Big( k_0^4 \big( V_x^2 V_x^6 - V_x^{4^2} \big) \big) + \Big( V_x^2 \big( 4k_0 c(3V_y^2 - 2c^2) $$
$$+ (6 + 5V_x^2 k_0^2)(V_y^2 - c^2) - 4k_0 V_y^3 $$
$$+ 2k_0^2 V_x^4 \big) - 16(c^2 - V_y^2)^2 \Big) $$

As $y \ll x$, the above quantity is above zero if

$$V_x^2 V_x^6 > V_x^4$$

which can be derived from Cauchy inequality. (25) is equivalent to

$$V_x^4 > V_x^2.$$

which can be derived from the Cauchy inequality.

$\square$

# D   More Results

**Visualization**   . We include more visualization comparisons. We show the comparison with generalizable methods and scene-specific methods in Fig. 6 and Fig. 7, respectively. We also include a failure case of CoFie in Fig. 8.

**Quantitative Results.**   We include a more comprehensive comparison with generalizable shape auto-encoding (GAE) and shape-specific auto-decoding (SSAD) methods for understanding the performance of our model. Again, we note CoFie performs generalizable shape auto-decoding (GAD) and is not directly comparable to these models.

Table 5: Performance on ShapeNet 10 novel categories. Specifically, the reported 3DS2VS [54] and NKSR [15] are trained on the full set of the training categories. In contrast, the reported numebrs in the main paper use a subset of 1000 instances for training.

| Setting | Method | CD (1e-4) | gIoU |
|---------|--------|-----------|------|
| GAD | DeepSDF | 10.4 | 83.1 |
| GAD | DeepLS | 7.27 | 96.2 |
| GAD | CoFie | 3.18 | 98.3 |
| GAE | 3DS2VS (full set) | 9.30 | 94.8 |
| GAE | NKSR (full set) | 4.24 | 96.9 |

Table 6: Performance on Thingi shapes. Note that SSAD methods take a long time for inference, e.g. NGLOD and UODFs take 105 and 300 minutes, respectively. In contrast, CoFie takes 10 minutes.

| Setting | Method | CD (1e-4) | gIoU |
|---------|--------|-----------|------|
| GAD | DeepSDF | 9.79 | 87.1 |
| GAD | DeepLS | 3.68 | 97.4 |
| GAD | CoFie | 1.87 | 99.0 |
| SSAD | NGLOD | 1.04 | 99.3 |
| SSAD | UODFs | 0.932 | 99.4 |

DeepSDF          DeepLS          Ours          GT

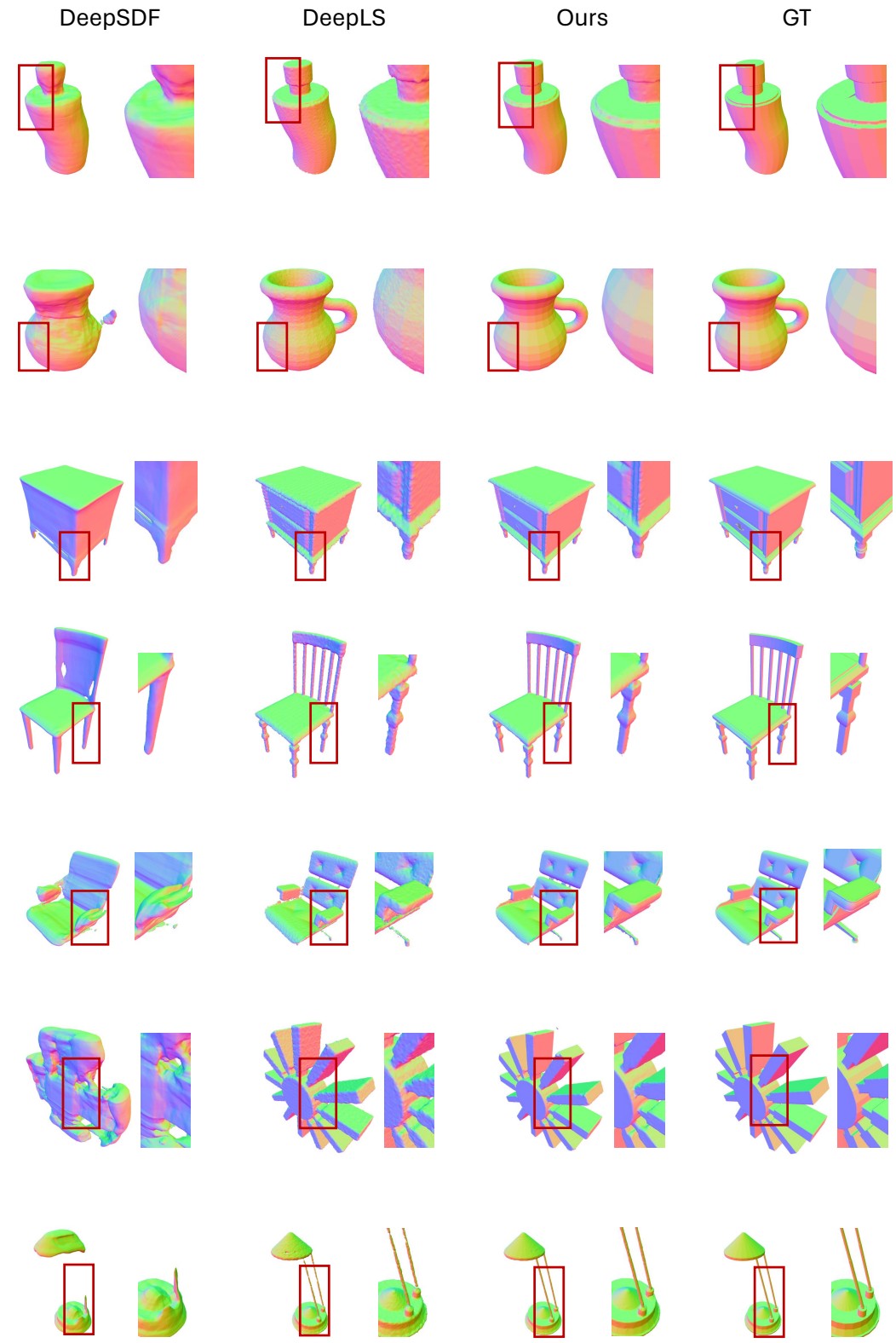

Figure 6: Compare with the generalizable methods DeepSDF and DeepLS on ShapeNet shapes. We show two images for each method, one for the overall shape quality, and a zoom-in detail check.

Ours             NGLOD            GT

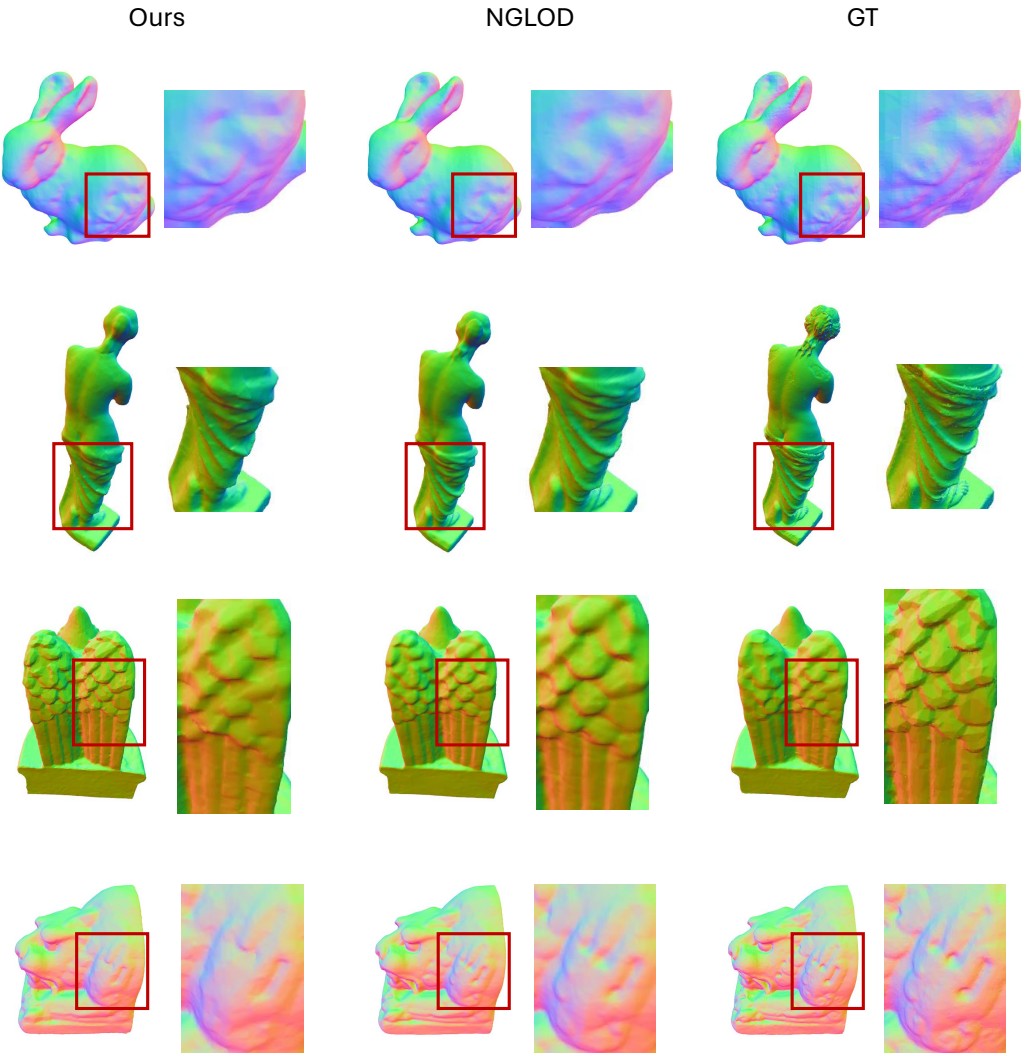

Figure 7: Compare with the shape-specific method NGLOD on Thingi shapes. We show two images for each method, one for the overall shape quality, and a zoom-in detail check.

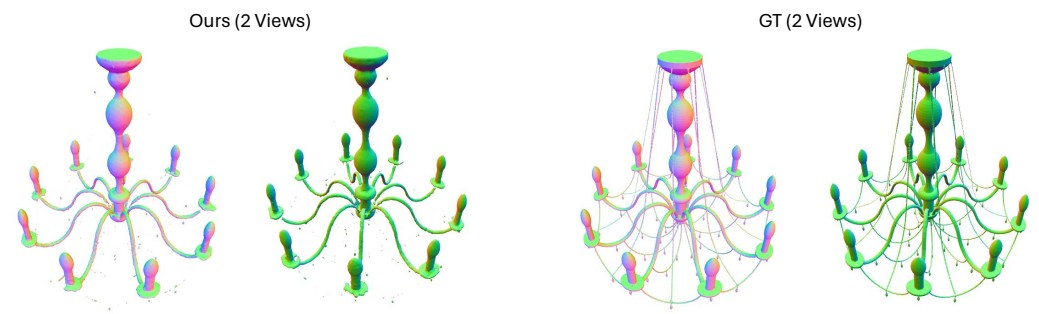

Figure 8: Analysis of the failure case. CoFie still struggles to represent extremely detailed geometry parts.

