# OpenReview forum: "CoFie: Learning Compact Neural Surface Representations with Coordinate Fields"
_NeurIPS.cc/2024/Conference — NeurIPS 2024 poster_

### Official Review · Reviewer_58nR · 2024-07-09

**Soundness:** 2
**Presentation:** 2
**Contribution:** 2
**Rating:** 3
**Confidence:** 5

**Summary:**

This paper proposes a new architecture for SDF auto-decoding task. It divides the whole SDF points into voxels and builds local coordinates for each surface patch included in valid voxels. Respective shape coding is learned separately and a generalizable MLP is used for SDF decoding. For MLP, a quadratic layer is proposed to better fit SDF values. Compared to the selected baselines, the proposed method shows great performance on both synthetic and real datasets.

**Strengths:**

The paper theoretically proves that ReLU based MLP is not enough for modelling the quadratic surface and thoroughly explains the difficulty in training local coordinate transformation. The results on two datasets show that local shape coding and hybrid representations have good potential for shape decoding.

**Weaknesses:**

The evaluation on the main assertions of this paper is not thorough. The method should be evaluated in three levels – accuracy, scalability and generalizability.

1. The first one has been partially demonstrated by comparing with the selected baselines. However, here a SOTA baseline - Neural Kernel Surface Reconstruction (CVPR’23) - is missing.

2. Scalability is not studied by the paper. Only object level datasets are tried. Large scale datasets such as CARLA should also be evaluated to demonstrate the ability of the model. This is a reasonable experiment for a locally coded model, because it only needs to introduce more and more patches for larger cases.

3. Generalizability is one of the selling points for this paper. However, there are some issues related to the baselines and experiment settings.

First of all, the paper states that the baseline NGLOD is a per-scene model, not a generalizable model, so NGLOD naturally could have better performance. However, NGLOD also has generalization ability according to the original paper, only LOD features are trained per-scene while keeping the MLP fixed, which is actually a similar setting to this paper.

Secondly, there are two kinds of generalizability: across domain and across density. In this paper only generalizability across domain, i.e. object level synthetic to real, is tried. Scene level, for example ShapeNet and Synthetic Room to ScanNet or Matterport3D, and generalization across density, that is trained and inferred on different sampling density, are missing.

4. Besides the experiments, there is one problem in theory derivation:

In the proof of proposition 1, equation 14 seems unreasonable to be equal. It is suggested to explain to what extend these two expressions could be regarded as equal. If equation 14 is not equal, how the final equation could be held?

5. Missing reference(s): Line 114, missing reference(s) for using quadratic surface patch

**Questions:**

Besides the weaknesses, there is one more question: currently all points in one voxel are assumed to be related to the surface within the same voxel, what if the points near the boundary of the voxel are related to the surface included in the nearby voxel?

**Limitations:**

Refer to weaknesses and questions.

---

> ### Author Rebuttal · Authors · 2024-08-07
>
> Thank you for the detailed feedback, your acknowledgment of our theoretical proofs of fitting quadratic local patches, and the potential of our method. We will address your concerns as follows.
>
> ---
>
> **[W1, Baseline]** Thank you for pointing us to NKSR. However, **NKSR is not a directly comparable method to ours**. As we mentioned in L189, L198, L247, and L249, our method works on the shape auto-decoding setting (the shape auto-decoder defined in DeepSDF), but NKSR works on shape auto-encoding. The differences are i) our method takes SDF samples as inputs, which is the same for DeepSDF and DeepLS, while the input of NKSR is the on-surface point cloud with normal as input; ii) After the shape decoder is trained, we fix it and perform back propagation to optimize the shape latent code, which is the same for DeepSDF, DeepLS and NGLOD, while NKSR uses feed-forward network to get the shape representation. Alternatively, as we mentioned in L248, even though the two settings are not directly comparable, we experimented with 3DS2VS, a SoTA shape auto-encoding method, as a reference method. To further address your concerns, we experiment with the mentioned NKSR method. We use 3K points as input without noise and use the full set of the 5 training ShapeNet categories for training; we test on the novel categories. We report the results on novel ShapeNet categories as follows. We note we also use the full set of 5 training ShapeNet categories for training 3DS2VS, while CoFie is trained on only 1000 shapes.
>
> |     | 3DS2VS | NKSR |  Ours  |
> |:---:|:-----------:|:-------:|:------:|
> | CD (1e-4)  |    9.32    |  4.24  | 3.18 |
>
> ---
>
> **[W2, Scalability]** Thanks for the valuable suggestions. We agree that CoFie can be extended to scene-level shapes as it uses local shapes. However, most of the scene meshes are not watertight, where we cannot use the point sampling method of DeepSDF to get the data. DeepLS paper provided a method to fake the input by using on-surface points and surface normal. However, their scene-level code is not released. We are unable to re-implement the scene-level data sampling method during the short rebuttal period, and we are committed to adding the result in the next version.
>
> ---
>
> **[W3.1, Generalization claim]** We thank you for the detailed feedback on generalization. However, the generalization in the NGLOD paper (reported in their Table 4) has a **different definition** from ours, even though we use the same terminology. As discussed at the beginning of Sec 4.3 of the NGLOD paper, their generalization works on the setting of i) training a model on **a single shape**, ii) applying the trained model to novel shapes, according to their description of “We now show that our surface extraction mechanism can generalize to multiple shapes, even from being trained on a single shape.” However, the generalization defined in our setting is i) training on **a set of shapes**, ii) applying the trained model to novel shapes, which is the setting in DeepSDF, DeepLS, and our work. We alternatively note that NGLOD doesn’t support training on multiple shapes. To further address your concern, we report the performance of NGLOD on Thingi by training the MLP on a ShapeNet Chair (denoted as NGLOD pre-trained). The performance is significantly worse than ours according to Table 3 in our paper. The reason is that the local shape distribution of the training shape and testing shape are very different. In contrast, our model can be trained on multiple shapes (1000 shapes with 3 million local surfaces), which enables us to learn general shape priors.
>
> |     | NGLOD (per-shape) | NGLOD (pre-trained) |  Ours  |
> |:---:|:-----------:|:-------:|:------:|
> | CD (1e-4)  |    1.04    | 3.68 | 1.87 |
>
> ---
>
>
> **[W3.2, Generalization across density]** We report the performance of our work using only 10% SDF samples as input. We observe that the performance of CoFie is robust to the number of SDF samples.
>
> |     | 100% SDF samples | 10% SDF samples  |
> |:---:|:-----------:|:-------:|
> | CD (1e-4)  |    1.87 | 2.64 |
>
> ---
>
> **[W4, Proof]**  Eq. 14 essentially applies Taylor expansion in x, y, z and ignores third-and-higher order terms in x, y, and z. Please see details in the rebuttal pdf.
>
> ---
>
> **[W5, Reference]** Thanks for the detailed feedback. The missing references at the mentioned line are [22, 38, 8, 34], which are the same with the ones we discussed in the second paragraph of the introduction. We will add them in the revision.
>
> ---
>
> **[W6, Shape at voxel boundary]** Good question! As mentioned in the paragraph of L211 of the paper, we deal with the problem by using an expanded “receptive field” for each voxel. In practice, we model a local shape using the SDF samples belonging to the neighbor voxel using a distance of 1.5V, where V is the size of the voxel. This improves the continuity of the shape at local voxel boundaries.

---

> > ### Comment · Reviewer_58nR · 2024-08-13
> >
> > Thanks for the very detailed rebuttal materials. I'm convinced that the proposed method is novel and could be potential to better represent 3D shapes. Nevertheless, my concerns regarding the three levels (accuracy, scalability and generalizability) of evaluation are just partially addressed. In the meantime, I also generally agree with reviewer V5fn that the paper needs more solid and fairer evaluations. Given the whole set of comments and newly added materials, I believe the current version clearly needs significant revisions to establish a decent baseline for future works, and therefore I would keep my negative score.

---

> > > ### Author Response · Authors · 2024-08-13
> > >
> > > Dear Reviewer 58nR,
> > >
> > > We have updated the discussion with reviewer V5fn, where we added GAE baselines trained with larger data (3DS2VS/NKSR with 20K shapes), and also added SoTA SSAD method UODFs. Wo hope these results could address your concerns on evaluation.
> > >
> > > ---
> > >
> > > For your comment
> > > > I believe the current version clearly needs significant revisions to establish a decent baseline for future works, and therefore I would keep my negative score.
> > >
> > > We believe our content doesn't need significant revisions. The truly important changes are just i) reporting results for 3DS2VS trained with 20K shapes, ii) reporting results for NKSR and UODFs, which are newly added baselines, iii) discussing these methods a bit more. All other content in the rebuttal and discussion are trying to resolving the questions from the reviewers.
> > >
> > > **Besides, we believe this is not a solid reason to keep the negative score just because we need to update our content. Otherwise, why we need this discussion and rebuttal period?** We hope the ACs could take this into consideration.
> > >
> > > ---
> > >
> > > Finally, we tried to resolve your concerns on accuracy (we provide NKSR results as you requested, and we also have other new baselines in the reply to other reviewers), scalability (we committed experiments on scene-level data in the future version), and generalizability (we clarified the different definition os generalization in NGLOD and our paper, which leads confusion to the reviewer). We hope these could be considered. The only one of your concerns on scalability is not directly addressed by experiments. At the same time, scalability to unbounded scenes is not a core property we focus in this paper, and we hope our contributions of i) theoretical analysis of local shape patches and ii) our coordinate field and its initialization could be considered.

---

### Official Review · Reviewer_Kq6M · 2024-07-10

**Soundness:** 4
**Presentation:** 3
**Contribution:** 4
**Rating:** 7
**Confidence:** 4

**Summary:**

### Motivation
- In the realm of implicit 3D shape representations, local-based solutions [4, 19, 32] (which decomposes the target shape into a set of local surfaces to model) provide higher accuracy but at the cost of a higher number of parameters to optimize.
- The authors argues that this number of parameters can be reduced by disentangling the actual geometry and the transformation (orientation, translation) of these local patches, modeling the latter separately.

### Contributions
- The authors thus introduce CoFie, a novel approach that utilizes an explicit coordinate field to transform local shapes, aligning them to reduce spatial complexity and improving MLP-based implicit local representation of 3D shapes.
- Various theoretical and empirical evidences are provided w.r.t. the benefits of their formulation, as well as to justify other technical choices, e.g., the usage of quadratic layers to complement their MLPs.

### Results
- Experiments show that CoFie reduces the shape error by 48% to 56% compared to traditional generalizable methods (e.g., DeepSF [4], DeepLS [24]) and achieves comparable performance to prior works with 70% fewer parameters.

### Relevance
- This paper addresses the challenge of balancing accuracy and compactness in neural surface representations, a relevant and generic issue in 3D shape modeling.

**Strengths:**

_(somewhat ordered from most to least important)_

### S1. Extensive Theoretical Grounding and Methodological Explanations
- The authors spent much effort formalizing and proving their contributions, providing the readers with extensive theoretical background.
- The Methodology itself is clear and thoroughly explained. While the actual implementation has not been provided, an expert in the art should be able to re-implement this work.
- The paper is also well illustrated and easy to follow (pipeline figures + qualitative results).

### S2. Novelty.
- To the best of my knowledge, the idea of disentangling geometry and transformation of local shapes is both new and interesting, and could benefit the 3DCV community.
- The authors nicely tie together their original intuition and extensive formalization to justify their contribution.

### S3. Convincing Evaluation
- The proposed CoFie method outperforms other generalizable methods in terms of shape accuracy and weight compactness.
- Qualitative results and the ablation study are also convincing.

**Weaknesses:**

### W1. Somewhat Limited Comparison to SOTA
- The authors compare to 4 methods (DeepSDF [24], DeepLS [4], NGLOD [30], 3DS2VS [41]), 2 of which are fairly old for the domain (<2020 for [24] and [4]). The literature has many more recent approaches, e.g. [a,b,c], with code or results pre-available, which could have been considered.
- Moreover, the authors compare to only one shape-specific method (i.e., performing test-time optimization), NGLOD [30]. While such methods tackle a slightly different task, comparing to a single solution makes it hard to draw solid conclusions.

### W2. Partial Evaluation of "Compactness" Claims
- The authors claim that "CoFie achieves comparable results with prior work using 70% less parameters" [L56-57]. However, there is a distinction to be made between parameter number (compactness of the model), convergence speed (number of iterations to convergence), and inference time / computational footprint. I find that claiming some improvement along one of those dimensions without studying the impact on the others not so meaningful.

### W3. Code Not Available
- While the authors claim that they are "committed to releasing our code" [L16], no code has been provided yet to reviewers.


### W4. Minor Remarks
- Citations are missing (empty brackets) in several places [L114, L128, L197].
- Typos: "practival: $\rightarrow$ "practical" [L118] ; "Perfomance" $\rightarrow$ "Performance" [L270] ; etc.


------------
### Additional References:

[a] Ye, Jianglong, et al. "Gifs: Neural implicit function for general shape representation." Proceedings of the IEEE/CVF Conference on Computer Vision and Pattern Recognition. 2022.

[b] Wang, Li, et al. "Hsdf: Hybrid sign and distance field for modeling surfaces with arbitrary topologies." Advances in Neural Information Processing Systems 35 (2022): 32172-32185.

[c] Lu, Yujie, et al. "Unsigned Orthogonal Distance Fields: An Accurate Neural Implicit Representation for Diverse 3D Shapes." Proceedings of the IEEE/CVF Conference on Computer Vision and Pattern Recognition. 2024.

**Questions:**

_see **Weaknesses** for key questions/suggestions._

### Q1. Computational Cost
- Besides the points made in **W2** above, I wonder what the computational cost of replacing linear layers with quadratic ones is (c.f. [L178-187])?

### Q2. Grid Size and Surface Complexity
- What is the impact of the voxel grid size on the surface complexity / ability of the model to learn the surface? Intuitively, the larger each voxel is, the more complex/discontinuous the surface that it intersects can become.

### Q3. Purpose of $w$
- What is the point of introducing $w$ [L168-171] if in practice it is set to the indicator function (simply selecting the voxel $v$ where $x$ belongs)? Is there a point to keeping Equation 5 this generic?

### Q4. Hyper-parameter $k$
- The authors mention that setting the number $k$ of quadratic layers (following $L-k$ linear ones) to 1 yields the best performance [L187]. Could the authors share the quantitative evaluation behind this conclusion?

**Limitations:**

- Limitations are discussed in the conclusion, with the authors sharing meaningful insight (e.g., the solution cannot perform shape completion or handle thin structures well).
- Societal impact are however not explicitly discussed. Providing some potential positive or negative applications of this work would be meaningful.

---

> ### Author Rebuttal · Authors · 2024-08-07
>
> Thank you for your detailed and positive comments. Thank you for acknowledging the novelty, the theoretical foundation, and the good performance of our work. We will address your comments as follows.
>
> ---
>
> **[W1, Baselines]** Although we understand your concern, we cannot find more directly comparable works. For example, GIFS, HSDF, and UODFs work on modeling non-watertight surfaces based on UDF. The input of GIFS and HSDF is the point cloud, which follows the setting of OccNet. However, the input of our method is SDF samples, where the points can be not on-surface. This follows the setting in DeepSDF, DeepLS and NGLOD. The mentioned UODFs work is a shape-specific method, which is directly comparable to NGLOD, but both NGLOD and UODFs are **different from our setting of generalizable shape representation as we noted in L244**. To further address your concerns, we report the performance of UODFs as follows on the Thingi mesh ID 252119. In general, UODFs demonstrate better performance than NGLOD. Our method, which takes much less time for inference and is not shape-specific, demonstrates comparable results.
>
> |     | NGLOD | UODFs |  Ours  |
> |:---:|:-----------:|:-------:|:------:|
> | CD  |    0.932    |  0.768  | 1.68 |
> | Inference Time | 105 min | 303 min | 10 min |
>
> ---
>
> **[W2, Compactness]** Thanks for your suggestion. Except for the number of parameters, our method also demonstrates a faster convergence speed over the directly comparable baseline DeepLS. When using the latent code size of 125, our method achieves comparable performance with DeepLS using only 61% training iterations. During inference, we set the same number of iterations for our method and DeepLS, where the inference time is roughly the same.
>
> ---
>
> **[W3, Code release]** We are firm supporters of open source. We commit to releasing the code right after the decision results.
>
> ---
>
> **[W4, Minors]** Thanks for the detailed feedback. We will incorporate them in the revision.
>
> ---
>
> **[Q1, Computational cost]** The quadratic layer makes the training/inference speed 14% slower. The used GPU memory increases by 3%.
>
> ---
>
> **[Q2, Grid size]** We experiment with a grid size of 16x16x16. The model shows a 4.96 chamfer distance error (according to Table 1 in the paper). We agree that the grid size influences the performance. In the meantime, we want to clarify that using a large grid resolution doesn’t increase the complexity of our model significantly. Our method works on valid voxels that intersect with the shape surfaces, which represent only 6% of all voxels when using a resolution of 32x32x32. This property makes our method efficient. When extending to a resolution of 64x64x64, only 1.2% of voxels are valid, where the complexity is only 1.6 times of resolution 32 and is far less than a cubical factor of 8.
>
> ---
>
> **[Q3, Aggregation function]** This function is a general representation of blending the local shapes into the global geometry. Using the indicator function is a special case of it, and is common in implementations, including DeepLS. Our method can also be extended to other blending methods, however, the blending method is not the focus and contribution of this paper.
>
> ---
>
> **[Q4, Hyper-parameter]** In our experiment, we also tested using 2 quadratic layers and the experiment showed a 2.20 error, which is larger than using one layer (according to Table 1 in the paper).

---

> > ### Comment · Reviewer_Kq6M · 2024-08-12
> >
> > I thank the authors for their thorough response, as well as my fellow reviewers for their insightful comments. I appreciate the authors' effort to address my concerns and questions. I am, therefore, still leaning towards acceptance, though I believe that Reviewer `V5fn` raised a few relevant points (e.g., SOTA selection, dataset size). I am following the ongoing discussion with attention.

---

> > > ### Author Response · Authors · 2024-08-12
> > >
> > > Thank you for the response and thanks for championing our paper. We will keep discussing with other reviewers.

---

### Official Review · Reviewer_4hsy · 2024-07-12

**Soundness:** 3
**Presentation:** 3
**Contribution:** 3
**Rating:** 6
**Confidence:** 3

**Summary:**

This paper introduces CoFie, a novel neural surface representation method designed to efficiently learn and represent complex 3D shapes. CoFie addresses the challenges of existing methods by introducing a coordinate field that optimizes the representation of local shapes, significantly reducing shape errors and improving the efficiency of parameter usage. The method is based on a hierarchical representation with coarse and fine-grained geometry, utilizing MLP with quadratic layers to enhance the expressiveness of the model. Experimental results demonstrate CoFie's strong generalization capabilities in shape reconstruction tasks, outperforming previous methods in terms of accuracy and efficiency.

**Strengths:**

CoFie uses a coordinate field to transform local shapes into an aligned coordinate system, reducing spatial complexity and making the learning of MLP-based implicit representations more efficient. By incorporating quadratic layers into the MLP, CoFie enhances the model's ability to capture local shape geometry, improving the quality of shape modeling. CoFie is trained on a curated dataset and can represent arbitrary shapes from novel categories, demonstrating strong generalization capabilities.
The theoretical proofs in the paper are sufficient and the articulation is clear.

**Weaknesses:**

1.CoFie's performance in detailing is strongly related to the cell size, which hinders its ability to represent fine-grained details.
2.Representations based on nonoverlapping local patches may not favor expressing rich colors, a capability that implicit representations possess.
3.It excels in displaying regular shapes, such as furniture, but is weak in showcasing shapes with rich curves.
4. Line 197, citation is missing

**Questions:**

I noticed that the article mentions, "We train CoFie on 1000 shape instances sampled from ShapeNet [5] consisting of chairs, planes, tables, lamps, and sofas (200 instances for each category)." Why not train on the entire ShapeNet?

**Limitations:**

Yes

---

> ### Author Rebuttal · Authors · 2024-08-07
>
> Thanks for your positive comments, and your acknowledgment that our theoretical proof is sufficient and the articulation is clear. We will address your comments as follows.
>
> ---
>
> **[W1, grid size]** Our method has demonstrated good performance in modeling detailed geometry as shown in Figure 3, Figure 6, and Figure 7, including thin structures, non-smooth local surfaces, as well as edges and corners. Alternatively, we note that our method can be easily extended to high resolution without increasing the complexity a lot. Our method works on valid voxels that intersect with the shape surfaces, which represent only 6% of all voxels when using a resolution of 32x32x32. This property makes our method efficient. When extending to a resolution of 64x64x64, only 1.2% of voxels are valid, where the complexity is only 1.6 times of resolution 32 and is far less than a cubical factor of 8.
>
> ---
>
> **[W2, Representing color]** Since we follow the common setting in implicit representation works [4,24,30,31,29,3,27,49,37], representing color is currently out of the scope of our work. Moreover, as shown in Figure 3, Figure 6, and Figure 7, our method can represent the local shape boundary between voxels accurately. This is strong evidence that it can also work for color. Moreover, as mentioned in L211, we improve the smoothness of the surface across different voxels by expanding the "receptive filed" if each voxel. This can also contribute to better color modeling capability.
>
> ---
>
> **[W3, Representing rich curves]** We kindly note that the reported number on Thingi is from the model trained on ShapeNet. Thus, its comparable performance with the shape-specific NGLOD demonstrates the power of CoFie on complex shapes.
>
> ---
>
> **[W4, Reference]** Thanks for the detailed feedback. We will revise the reference.
>
> ---
>
> **[Q1, Training set]** Training on 1000 shapes of ShapeNet is a common setting in shape auto-decoding works. We strictly follow the setting in DeepLS for using 1000 shapes. Please see more details in the experimental setup section of the DeepLS paper. Besides, we want to clarify that the 1000 shapes contain 3 million local surface samples, which is sufficient for training. This property demonstrates the data-efficient nature of our work.

---

> > ### Comment · Reviewer_4hsy · 2024-08-12
> >
> > I have read the authors' rebuttal and appreciate their efforts during the rebuttal process. Therefore, I maintain my initial rating.

---

> > > ### Author Response · Authors · 2024-08-12
> > >
> > > Thank you for your acknowledgment of the effectiveness of our rebuttal. We will incorporate the discussion in revision!

---

### Official Review · Reviewer_V5fn · 2024-07-13

**Soundness:** 2
**Presentation:** 2
**Contribution:** 2
**Rating:** 3
**Confidence:** 4

**Summary:**

Paper proposes CoFie — 3D shape representation as a set of latents arranged on a regular voxel grid. Each latent encodes an oriented local quadratic patch. This local oriented patch defines local SDF in a local coordinate frame which is decoded via conditional MLP for which the last layer is quadratic: it defines a bilnear form on input vector instead of linear mapping). SDF value for query point x is decided based on the SDF value of local implicit function belonging to the nearest voxel and transformed from local coordinate frame to global.

Authors train proposed methods and baselines on a single random subsample of 1000 ShapeNet shapes (5 categories, 200 shapes each) and evaluate on holdout ShapeNet shapes and out-of-distribution shapes from Thingi10K. Proposed method outperforms baselines trained in the same limited data regime. The same ShapeNet model is also evaluated with the Neural Geometric Level of Detail method and DeepSDF fitted in the shape overfitting scenario.

**Strengths:**

— Proposed representation is more efficient compared to regular latent grids and is comparable to surface latent grids (3DILG) in terms of latents needed to represent shape;
— Method figure is clear and well done and quantitative results are well formatted;
— Qualitative results look comparable to NGLOD which is a strong baseline for shape overfitting setting;
— Design choices are clearly ablated (Table 4);

**Weaknesses:**

— Proposed method seems to be very similar to AutoSDF and SDFusion: both methods utilize a regular latent grid that encodes local SDF that is used to infer the global SDF. Main difference seems to be in a local SDF decoder that is quadratic (last layer) for proposed method and also encodes SDF in the local coordinate frame (SDFusion and AutoSDF use global coordinates). Overall, it is fine as long as proposed method compares to these similar methods but evaluation of the paper is limited (see below).

— Similarly to other regular latent grids methods (e.g. SDFusion, AutoSDF), this model scales cubically with respect to grid resolution and thus might not be well suited for representation of topologically challenging shapes with thin parts.

— Evaluation of the paper is extremely limited. For point cloud reconstruction (auto-encoding), proposed methods and baselines are trained on 5 ShapeNet categories with 200 shapes per category (overall 1000 shapes). This evaluation is not enough to support the claim that CoFie “can represent arbitrary shapes that belong to any novel category” (LL52-53). These results might only indicate that CoFie might be better representation in low-data regime but in this case evaluation should be done similar to few-shot learning setting: all models should be trained on several different subsets of ShapeNet (e.g. 5) and average/std of test errors should be reported. I also want to note that 3D2VS trained on full ShapeNet is an extremely strong baseline that achieves almost perfect reconstruction quality for some categories (like airplanes). Also, the paper only uses Chamfer distance for evaluation and ignores other common measures like IoU and surface F-Score.


— The choice of baselines is also limited. DeepSDF is a 2019 paper that uses a very simple Pointnet (not even PointNet++) encoder. DeepLS is a 2020 paper. 3D2VS seems to be the only recent baseline for 3D point cloud reconstruction. It is a very strong baseline but authors have not used pretrained models and trained their version on limited data instead. Given the fact that 3D2VS uses a very high capacity attention-based decoder, this training regime might not be a fair comparison because attention-based models often struggle in low data regimes. For the same reason, 3D2VS might not be a good baseline if the goal of the paper is to show that their representation is compact. In this case, 3DILG might be a more suitable and strong baseline (see below) because it uses a simpler local patch encoder (Pointnet), so it might work better in a limited data regime. Since the model utilizes local surface patches similarly to AtlasNet (see links below), it also can be a strong baseline, especially in low data setting.

— Evaluation in an overfitting setting  does not seem methodologically correct. : NGLOD and DeepSDF overfit to single shapes while CoFie was pretrained trained on 1000 shapes. These are completely different settings: overfitting regime measures capability of the model to efficiently fit one shape with preservation of geometric detail, and fitting and evaluation on collection of shapes evaluates the ability of the model to generalize to unseen shapes. This experiment should either be done if fully overfitting setting (similar to NGLOD paper) or in a reconstruction setting (see concerns above).

— Quadratic MLP contribution seems a little bit weak to me. It looks like it is basically equivalent to linear MLP being run on quadratic feature expansion and in this case commonly fourier embeddings of points can be a very strong alternative (this is not tested).

— Paper writing can be significantly improved in clarity. For example, paper does not specify input to auto-encoder for baselines. Is it point cloud or mesh? If it is a point cloud, what is the density of sampling and what sampling was used (FPS, random, poisson disk sampling, etc). How many codes were used to train 3D2VS? 512? What were hyperparameters for baseline training? On the other hand, paper spends a lot of space describing relatively common knowledge like MLP with ReLU activations (LL172-177). Another example is equation (5). On first glance it appears as weighted average, but since authors use voxel indicator function as weight, it means that for each query point x, global implicit function is based only on implicit function of nearest voxel grid latent.

— Some statements in the paper are not correct. For example, the paper states that 3D2VS “employs transformers to predict the shape latent code” (LL258-260) . This is not factually correct: 3D2VS does not represent shape as one vector, it uses latent code clouds (usually 512 codes per shape).

— Representation seems to be computationally expensive. Training on 1000 shapes takes one day on 4 GPUs with 24 GB VRAM.

— Some relevant work is missing

AutoSDF: Shape Priors for 3D Completion, Reconstruction and Generation https://arxiv.org/abs/2203.09516

SDFusion: Multimodal 3D Shape Completion, Reconstruction, and Generation
https://arxiv.org/abs/2212.04493

4D Spatio-Temporal ConvNets: Minkowski Convolutional Neural Networks https://arxiv.org/abs/1904.08755

3DILG: Irregular Latent Grids for 3D Generative Modeling https://arxiv.org/abs/2205.13914


AtlasNet: A Papier-Mâché Approach to Learning 3D Surface Generation https://arxiv.org/abs/1802.05384

**Questions:**

— What is the input for the model for auto-encoding settings? Is it a point cloud? Or model back propagates into learnable latent code like DeepSDF? If it is a point cloud, what encoder model is used? If model back propagates into latent code, how come it is generalizable?

— How were baselines trained? Were they trained on the same inputs as the proposed method? Did all methods use the same SDF sampling? As far as I know, 3D2VS uses occupancy as supervision. Was it trained with SDF or occupancy supervision for the paper?

— What motivated size 1000 for a training dataset? Majority of modern shape auto-encoder train either on full ShapeNet of full ShapeNet categories (e.g. ~6000 ShapeNet chairs). If 5 different subsets of ShapeNet of size 1000 would be selected (same categories), how high would be the variance in evaluation across these subsets? I have strong suspicion that results shown int Tables 1 and Tables 2 might not generalize across training subsamples.

— Have you tried using Fourier embeddings of input coordinates instead of quadratic layers? This might be a very strong alternative since it helps coordinate based models to encode high-frequency details very efficiently.

**Limitations:**

— Proposed method represents the shape as a set of latents arranged on a regular latent grid and thus scales cubically with latent grid resolution. This limits method’s ability to efficiently encode topologically challenging shapes.

— Method requires a lot of compute to fit 1000 shapes (1 day on 4GPUs with 24GB VRAM). This might be due to fact that authors use 32^3 latent grid (SDFusion for example uses 8^3). This severely limits scalability of the proposed method even for training on full ShapeNet, not to mention larger datasets like Objaverse.

---

> ### Author Rebuttal · Authors · 2024-08-07
>
> Thanks for your detailed feedback, and your acknowledgement of the efficiency of our model, the clarity of our writing and figure, and our clear ablation study. We will address your comments and some **factual errors** in your comments as follows.
>
> ---
>
> **[W1, AutoSDF and SDFusion]** The mentioned papers are significantly different from our work. First, AutoSDF and SDFusion have a different target from ours. In detail, AutoSDF and SDFusion target to learn the latent space of SDF represented shape, which enables them to perform downstream applications of shapes that belong to specific categories, e.g., chairs and airplanes. In contrast, our work focuses on shape representation, where we prove that using coordinate fields with better MLP designs can represent the shape of general categories better with both experimental and theoretical evidence. Moreover, AutoSDF and SDFusion use shape auto-encoding to learn the latent space. Differently, our paper works on shape auto-decoding (as mentioned in L189 and L243), or termed as auto-decoder in DeepSDF and DeepLS paper. As discussed in L248, shape auto-encoding works can not be directly compared with our method. **At the same time, we agree with your comment that we both use grid-based SDF. However, this is very common and standard in related research and we have discussed them in Sec.2, including [4,21,36,19,32,35,30]. We never claim that using the grid-based SDF is one of our novelties.** We are happy to add the discussion of the mentioned paper in the revision for more clarity.
>
> ---
>
> **[W2, Complexity]** **Our method doesn’t scale with cubical complexity.** As illustrated in Fig 2, our representation only focuses on grids that contain the shape surface. Thus, we actually use **sparse voxels**, and those voxels that don't contain surfaces are discarded. When using the resolution of 32x32x32, only 6% percent of the grids are valid. When using the resolution of 64x64x64, only 1.2% of voxels are valid, where the complexity is only 1.6 times of resolution 32 and is far less than a cubical factor of 8. This property is similar to DeepLS.
>
> ---
>
> **[W3, Evaluation]** The effectiveness of our evaluation is acknowledged by other reviewers, including “convincing evaluation” (Kq6M), “good ablation study” (LpYZ), “Experimental results demonstrate strong generalization capability of CoFie, outperforming previous methods in terms of accuracy and efficiency”, and “the results on two datasets show great potential” (58nR).
>
> To further address your comments, first, we would kindly remind you that CoFie works on the shape auto-decoding task (optimization-based) as we mentioned in L189, L198, L247, and L249. CoFie doesn’t work on the auto-encoding task (feed-forward-based) you mentioned.
> Second, regarding our claim of “represent arbitrary shape”, we have verified the claim in our experiment in Sec 5, and it is appreciated by other reviewers, including “strong generalization capability” (4hsy), “outperform previous generalizable methods” (Kq6M), and “show great performance on both synthetic and real data” (58nR). As we mentioned in L277, we train the model on 5 ShapeNet categories, and we test the model on 10 novel ShapeNet categories and also novel real objects from novel categories. Results in Table 2 and Table 3 are comparable to Table 1, demonstrating the strong generalization capability. Conceptually, our method doesn’t have any category-specific designs, and the local patch-based method is naturally generalizable to any shape [4].
>
> Finally, we would like to address your concerns regarding the dataset size. First, using a thousand instances for training is typical in previous works [24,4]. We strictly follow the setting in DeepLS [4]. Second, to further address your concern, we report the result of 3DShape2Vec using the full set of the 5 categories in Table 1 below. We alternatively mention that 3DS2VS is a shape auto-encoding method and it is not directly comparable to shape auto-decoding methods including DeepSDF, DeepLS, NGLOD, and our method. We have mentioned this in L248. Shape auto-decoding generally has a much better performance than shape auto-encoding. With more data, the shape representation accuracy of 3DS2VS is still not as good as shape auto-decoding methods, especially in novel categories.
>
> ---
>
> **[W4, baselines]** We respectfully disagree with your points. **You mentioned that DeepSDF uses a PointNet encoder, but actually there is no encoder in DeepSDF as it performs shape auto-decoding (or termed as auto-decoder). By searching “PointNet” in the DeepSDF paper, we don’t find any matches in the method section.** Please let us know if there is any following work that modifies DeepSDF with a PointNet encoder. This is also true for DeepLS. The mentioned 3DILG paper works on shape auto-encoding, and actually the SoTA of shape auto-encoding is 3DS2VS and the performance of 3DS2VS is reported in Table 3 of 3DS2VS paper, which outperforms 3DILG. Thus, we didn’t compare with 3DILG. Again, we want to emphasize that shape auto-decoding methods and auto-encoding methods are not directly comparable. We would be happy to compare with more shape auto-decoding methods if we missed any.
>
> |     | 3DS2VS (1000 shapes) | 3DS2VS (full set) |  Ours  |
> |:---:|:-----------:|:-------:|:------:|
> | CD  |    16.0     |  9.3 | **3.2** |
>
> ---
>
> We will continue to address the question in the official comment reply.

---

> ### Author Response · Authors · 2024-08-07
>
> **[W5, Evaluation]**  **You mentioned that DeepSDF overfits single shapes, but in fact, it trains the MLP decoder on a set of training shapes, and optimizes the latent code for each shape during testing**. The MLP decoder is shared for all shapes, thus, it is not overfitting. Please see details in the DeepSDF paper and the training split of the DeepSDF official GitHub repo. We strictly follow this setting. We agree that NGLOD overfits each single shape (shape-specific) for both the MLP decoder and shape representation. As we discussed in L245, shape-specific methods generally have better accuracy as it is specialized in a single shape. We use it as an “oracle” comparison, and the results in Table 3 show the strong performance of our method.
>
> ---
>
> **[W6, Quadratic MLP]** The use of quadratic MLP is based on our analysis with a strong theoretical foundation (Sec 3).
>
> ---
>
> **[W7, Paper details]** The clarity of our writing is acknowledged by other reviewers, including “proofs are sufficient and articulation is clear” (4hsy) and “well illustrated and easy to follow” (Kq6M). As we mentioned in L201, the inputs are SDF samples, not point cloud or mesh. The density and method of SDF sampling strictly follow DeepSDF, which is also the same in our experiment of DeepLS and NGLOD. For 3DS2VS, we use 10K points as input for a fair comparison, as DeepSDF sampling has the number of SDF samples with this quantity.
>
> ---
>
> **[W8, Statements]** Shape latent code is the official terminology in 3DS2VS. See the second line of the caption of Figure 2 in 3DS2VS. “Latent code” doesn’t mean there is only one code. We searched in the 3DS2VS paper, and there is not a single matching regarding the term “latent code clouds” you mentioned.
>
> ---
>
> **[W9, Computation]** Our method is computationally efficient. The 1000 training shapes contain 3 million local patches, and we train on the patch level. When using the latent code size of 125, our method achieves comparable performance with DeepLS using only 61% training iterations. This is strong evidence that our method is computationally efficient.
>
> ---
>
> **[W10, Reference]** We are happy to cite and discuss the mentioned papers in the revised version of our paper.
>
> ---
>
> **[Q1, Input]** As we mentioned above, we are performing shape auto-decoding (updating latent code via back propagation and optimization during testing), where the standard input is SDF samples. The input of DeepSDF, DeepLS, and CoFie are strictly the same. We don’t have an encoder as we perform shape auto-decoding. Regarding generalization, there is no correlation between generalization and whether it uses back propagation. See DeepLS, which is a generalizable method using back propagation, for more details.
>
> ---
>
> **[Q2, Baselines]** As we mentioned above, DeepSDF, DeepLS, NGLOD, and our method use strictly the same inputs. We follow the 3DS2VS official code for training and testing.
>
> ---
>
> **[Q3, Training data size]** Again, we are doing shape auto-decoding, which is a different task from auto-encoding. We agree that shape auto-encoding requires a large amount of training data as it learns priors to map the shape/point cloud into latent code. However, this is not true for shape auto-decoding as these methods optimize the latent code during inference. For example, DeepLS also uses 1000 shapes for training (as discussed in the experiment setup paragraph in the DeepLS paper) and we follow their setting strictly. Our method works on a patch level, and the 1000 shapes introduce 3 million local patches, which are sufficient for training.
>
> ---
>
> **[Q4, Using Fourier embedding]** Good question! We experiment with using Fourier embeddings and show the results below. To make the comparison fair, we use a code size of 64 and Fourier embedding with a dimension of 64, which makes the input dimension the same as using point coordinates (3 dimensions) with a 125-dimensional latent code. We provide the results as follows, and they show the effectiveness of our method. Alternatively, we note that our idea of coordinate field is orthogonal to using positional encoding, as we can also apply positional encoding on the transformed coordinate system.
>
> |     | DeepLS| DeepLS + Fourier Embedding | Ours |
> |:---:|:-----------:|:-------:|:----:|
> | CD (1e-4)  |   7.27 | 6.22 | 3.18 |
>
> ---
>
> **[L1, Representing complex shapes]** We have already shown the capability of CoFie in Figure 3, Figure 6, and Figure 7, which include topologically challenging shapes like thin structures and unsmooth local surfaces, as well as edges and corners.
>
> ---
>
> **[L2, Computation]** Please see W2 and W9. Alternatively, we report that NGLOD takes 105 minutes for a single shape, which is much slower than ours.

---

> ### Comment · Reviewer_V5fn · 2024-08-12
> **Response to main rebuttal**
>
> I appreciate very detailed feedback by the authors: it clarified a lot of details in the original submission. Below I will address main rebuttal points (and I will address other rebuttal points in separate comments).
>
> After the rebuttal, I have no doubts that Cofie proposes a novel method that can be viewed as a sparse irregular latent cloud/grid/array of 1/32 sized voxel cells that intersect with shape surface. These cells are decoded via novel quadratic MLP decoder (and this choice is ablated in the rebuttal).  This makes it different from 3DILG, Dynamic Code Clouds, 3DS2VS, AutoSDF and SDFusion.
>
> Unfortunately, my three major concerns about evaluation are not addressed in the rebuttal.
>
> **Concern 1. Choice of baselines for auto-decoding setting (Tab. 1) is very weak.**
>
> DeepSDF is the 2019 paper. DeepLS is a 2020 paper. Since then, a lot of strong shape representations were proposed: Local Implicit Grids (2020), AutoSDF (2022), SDFusion (2023), Dynamic Code Clouds (2023), 3DILG (2022). I don’t expect authors to compare to all of them but in addition to 3DS2VS proposed method needs comparison with 2-3 recent methods that utilize latent grids/clouds/arrays.
>
> **Concern 2. Proposed method is evaluated via training on 1000 ShapeNet shapes (Table 1).**
>
> Authors claim that “using a thousand instances for training is typical in previous works [24,4]”. Again, DeepSDF [24] is the 2019 paper; DeepLS [4] is the 2020 paper. For comparison, 3DS2VS trains on 48597 ShapeNet shapes (auto-encoding task) and evaluates on 2592 shapes. In fact, recent works use at least ShapeNet-13 (see Dynamic Code Clouds papers). In my opinion, the proposed method must either be trained and evaluated at least on ShapeNet-13 (as Dynamic Code Clouds does) or results for 1000 shapes must be averaged across at least 5 different runs on different 1000-shape subsamples of ShapeNet.
>
> **Concern 3. The paper only uses Chamfer Distance as an evaluation metric.**
>
> Majority of baselines use at least 2 evaluation metrics. DeepSDF [4] uses CD and EMD; 3DS2VS uses CD, IOU and F-score. Only DeepLS uses just a CD. Given the strong claims in the paper and limited training/test size in the submission, I recommend having at least one additional evaluation metric.
>
> *If the paper with current evaluation pipeline is accepted, my concern that it might create bad precedent for follow-up work that is going to use the same evaluation pipeline. Thus, my rating remains unchanged.*
>
> ======================================
>
> Below are additional main rebuttal points I want do address:
>
> > By searching “PointNet” in the DeepSDF paper, we don’t find any matches in the method section
>
> Authors are right and I acknowledge my mistake and apologize for the confusion. But I want to point out that this mistake does not invalidate my concerns outlined above and the fact that these concerns were not sufficiently addressed in the rebuttal.
>
> > The effectiveness of our evaluation is acknowledged by other reviewers
>
> The goal of the review process is to gather unbiased and diverse feedback from fellow researchers, so it is only natural that opinions differ.

---

> ### Comment · Reviewer_V5fn · 2024-08-12
> **Response to comments #2**
>
> >  We searched in the 3DS2VS paper, and there is not a single matching regarding the term “latent code clouds” you mentioned.
> > Shape latent code is the official terminology in 3DS2VS.
>
> I refer authors to the 3DShape2VecSet paper introduction and contribution 1 in particular: "We propose a new representation for 3D shapes. Any shape can be represented by a fixed-length array of latents".  I also refer authors to section 5.1 to 3DS2VS paper that explicitly states: "In our final model, the number of latents 𝑀 is set as 512, and the number of channels 𝐶 is 512". Finally, I refer authors to the [source code of 3DS2VS (LL182-283)](https://github.com/1zb/3DShape2VecSet/blob/master/models_ae.py).
>
> >  “Latent code” doesn’t mean there is only one code
>
> I don't believe authors are making this argument in good faith. DeepSDF paper explicitly states that shape is represented by 1xD vector. 3DS2VS paper explicitly states that shape is represented by 512xD tensor (see above). In scientific research, precision of language matters. For example, nobody call 'matrix' (2D tensor) a 'vector' (1D or 0D tensor).
>
> > The use of quadratic MLP is based on our analysis with a strong theoretical foundation (Sec 3)
>
> The argument in Sec. 3 is done in comparison to 'shallow MLP using ReLU'. This assumption does not hold in majority of practical cases: majority of methods either use non-shallow MLPs, or non-locally linear activations like (ELU, GEGLU, GELU, SiLU, etc), or both.
>
> > We experiment with using Fourier embeddings and show the results below <...>
>
> Thank you! This is solid empirical evidence in favor of proposed method and I recommend including it in updated version of the paper.
>
> >  Our method works on a patch level, and the 1000 shapes introduce 3 million local patches, which are sufficient for training.
>
> I respectfully disagree. The fact that models sees 3 mln local patches of 1000 ShapeNet shapes during training does not mean that it will be able to generalize to 50000 unseen ShapeNet shapes. I also want to note that 3DILG also utilizes local surface point patches (512 per shape) that are randomly sampled for each iteration. It means that for each iteration of 1000 shapes 3DILG sees ~0.5 mln of unique local patches, so this property is not unique to the proposed method.
>
> > Ability to deal with topologically challenging shapes
>
> I respectfully disagree. Maybe the issue here is that we that we consider different types of shapes topologically challenging. I refer authors to Fig. 8 of 3DS2VS paper for examples of topologically challenging shapes that I find topologically challenging (esp. rows 2,3,5,8,9). My concern is that proposed method won't be able to handle such shapes (and 3DS2VS is capable of it).
>
> >Our method is computationally efficient <...>.
>
> I respectfully disagree. While proposed method might be more computationally efficient than DeepLS (2020 paper), it still requires 4 24GB GPUs and 24 hours to train on 1000 shapes. To me, it means that it will take at least a month to train this model on full ShapeNet. But given more pressing issues with evaluation and baselines, this does not hold high weight in my evaluation.
>
> > You mentioned that DeepSDF overfits single shapes, but in fact, it trains the MLP decoder on a set of training shapes, and optimizes the latent code for each shape during testing.
>
> My confusion is understandable since proposed method compares both to NGLOD that was trained in overfitting setting and 3DS2VS that is trained in auto-encoding setting. That being said, I apologize for the confusion but want to point out that this confusion does not invalidate my concerns.
>
> > The clarity of our writing is acknowledged by other reviewers <...>
>
> The goal of review process is to gather unbiased feedback from fellow researchers, so it is only natural that opinions differ.

---

> > ### Author Response · Authors · 2024-08-12
> >
> > We thank the reviewer for the detailed reply and we appreciate the efforts! We are happy that we have resolved the factual errors in the original rebuttal. We will address the new comments as follows.
> >
> > ---
> >
> > **[C1, Baselines]** We understand your concern regarding the baseline. However, as we noted in our rebuttal, **we didn’t miss any directly comparable baselines** (generalizable shape auto-decoding) as far as we know. **The suggested baselines work on different tasks** of generalizable shape auto-encoding or shape-specific auto-decoding, where the train-test paradigm and inputs are different from our setting. **We have already included the representative works for the two tasks**, NGLOD (2021) and 3DS2VS (2023), **as reference in our paper to understand the performance of CoFie**.
> >
> > To further address your concerns, we would like to point to the **two new baselines we added during the rebuttal period** (in the reply to other reviewers). We include the results of **NKSR** (2023, generalizable shape auto-encoding with additional point normal input) and **UODFs** (2024, shape-specific auto-decoding) in Tables R1 and Table R2. We report the results of NKSR on the 10 novel ShapeNet categories, and report the results of UODFs on the Thingi shapes (due to its slow inference speed of 300 minutes per shape).
> >
> > We note that directly comparing the numbers across different settings is not meaningful. Instead, we should compare CoFie with methods in the same setting (DeepSDF, DeepLS). **We agree that DeepSDF and DeeepLS are relatively old works, but this is perpendicular to our contributions in this work, as every contribution is ablated**.
> >
> > ---
> >
> > **[C2, Training data size]** In general, we agree that recent works are using larger data for training. At the same time, we would like to provide more explanation why we didn’t use super large-scale data for training. We note that **the data size is closely related to the task and model capacity**. *The recent generalizable shape auto-encoding methods use large encoder networks, where larger data is necessary to train those large models. In contrast, shape auto-decoding methods typically use shallow MLPs (DeepSDF, DeepLS, NGLOD, etc), where the benefit of using larger data is smaller*. Again, we note that **we strictly follow the training data scale of prior directly comparable works**, i.e., DeepSDF and DeepLS.
> >
> >
> > Moreover, the reviewer questioned our training efficiency, e.g. 1000 shapes takes 1 day for training so using larger data takes longer training time linearly. We would like to clarify that we set a total iteration number for training, e.g. 150K iterations (in each iteration we train with 36K local patches), which is sufficient. When using larger data, we won’t linearly scale the training iterations.
> >
> > Besides, the authors mentioned recent shape auto-encoding methods using **ShapeNet-13** for training/testing and requested our method to perform experiments on the same amount of data/categories. In response, we would like to remind you that **we perform experiments on ShapeNet-15 categories** (5 categories for training, and 15 categories for testing). The performance on the 10 novel testing categories strongly suggests that CoFie can generalize to novel categories.
> >
> > Finally, we are happy to provide experimental results using more data (full set of ShapeNet-5 categories) for training. Additionally, we note that **this experiment is not explicitly required in the first review comment**. We will try our best to finish the experiment within the discussion period.
> >
> > ---
> >
> > **[C3, More Evaluation Metric]** We thank the reviewer for the suggestion! We report the gIoU metric following NGLOD as follows.
> >
> > **Table R1. Performance on ShapeNet 10 novel categories.** GAD denotes Generalizable Shape Auto-decoding, and GAE denotes Generalizable Shape Auto-encoding.
> >
> > | Setting | GAD        | GAD        | GAD        | GAE               | GAE               |
> > |---------|------------|------------|------------|-------------------|-------------------|
> > | Method  | DeepSDF    | DeepLS     | CoFie      | 3DS2VS (full set) | NKSR (full set)   |
> > | CD (1e-4)| 10.4      | 7.27       | 3.18       | 9.30              | 4.24              |
> > | gIoU     | 83.1       | 96.2       | 98.3       | 94.8              | 96.9              |
> >
> > ---
> >
> > **Table R2. Performance on Thingi shapes of arbitrary categories.** SSAD denotes shape-specific auto-decoding.
> >
> > | Setting | GAD        | GAD        | GAD        | SSAD              | SSAD              |
> > |---------|------------|------------|------------|-------------------|-------------------|
> > | Method  | DeepSDF    | DeepLS     | CoFie      | NGLOD             | UODFs             |
> > | CD (1e-4)| 9.79      | 3.68       | 1.87       | 1.04              | 0.932             |
> > | gIoU     | 87.1       | 97.4       | 99.0       | 99.3              | 99.4              |
> >
> > Additionally, we noticed an error in our Table 3 in the paper, where results for DeepSDF should be for DeepLS.

---

> > > ### Author Response · Authors · 2024-08-12
> > >
> > > We will continue to reply to the comments.
> > >
> > > ---
> > >
> > > - **3DS2VS terminology.** We realized that “latent code” can lead to misunderstanding from the clarification of the reviewer. We will revise it to “latent code set” as it titled. In the experiment, we use a set of 512 codes so no worries about it. Besides, it seems that the material you provided also doesn’t contain “latent code clouds” you mentioned, but we realize this may not be a meaningful battle.
> > > - **Quadratic MLP**. We agree that this is studied in the context of shallow MLP with ReLU and we kindly point the reviewer to DeepSDF, DeepLS, NGLOD where the context is adopted and we directly compare CoFie with them. We agree that using better activations or deeper MLP can improve the performance, but this is perpendicular to our contribution.
> > > - **Patch-level property**. Please see our reply to your concern 2 on why using small-scale data for training. CoFie sees 36K local patches in one iteration. Our original reply was explaining why 1000 shapes can be sufficient and we never claim this is unique to CoFie. DeepLS has the same property.
> > > - **Challenging shapes**. Thanks for the clarification. We will incorporate more visualization in these cases. We kindly refer the reviewer to Figure 8 in the appendix, where CoFie fails on super challenging thin structures, and we believe this is a common problem for any shape representation.
> > > - **Computational efficiency**. Please see our reply to your concern 2 for more details.
> > >
> > >  ---
> > >
> > > Finally, we would like to **reiterate our contributions**. Generally, we intend to **inject geometry-aware inductive bias for shape representation**. Thus, we i) provide theoretical analysis on the quadratic shape patch properties, and propose to ii) disentangle the geometry and transformation information of local patches by using geometry-aware coordinate fields and initialization method, and iii) leverage quadratic operations to improve the representation capability on quadratic local patches. These contributions make our shape representation easier to learn and compact (achieving same performance using a much smaller code size), and the contributions are ablated. **We would appreciate it if these contexts of larger goals/contributions could be considered.**
> > >
> > > Again, we would thank the reviewer for the detailed feedback. We will discuss the mentioned works in the revision and incoporate the discussions. We sincerely hope this reply can be considered.

---

> > > > ### Comment · Reviewer_V5fn · 2024-08-12
> > > >
> > > > Again, I appreciate the effort authors are putting in the rebuttal.
> > > >
> > > > I want to reiterate that I have no doubts about novelty of the proposed method. My main concern is **weak evaluation** of the proposed method. The proposed method either compares to very old (by field's standards) methods or compares to very strong recent methods in a setting that might be unfavorable to them like training on 1000 shapes. When the model compares to strong baselines in fair setting (see Thingi10K results) the proposed method does not outperform them (again, see Thingi10K results).
> > > >
> > > > Minor comments below:
> > > >
> > > > > 3DS2VS terminology.
> > > >
> > > > I appreciate authors acknowledging that usage of 'latent code' with respect to 3DS2VS representation can lead to ambiguity, especially for readers who are less familiar with the recent advances in shape representation via latent grids/arrays/clouds.
> > > >
> > > > > We agree that this is studied in the context of shallow MLP with ReLU and we kindly point the reviewer to DeepSDF, DeepLS, NGLOD where the context is adopted and we directly compare CoFie with them.
> > > >
> > > > This again brings us to the question whether the choice of the baselines of the paper is strong. As I mentioned, a lot of recent methods don't use ReLU and don't even use MLPs that much (for example 3DS2VS uses GEGLU in attention modules that are utilized both in encoder and decoder modules). My point is that 'strong theoretical foundation' for the paper is justified in very limited setting that does not hold in vast majority of practical cases.
> > > >
> > > > >  We kindly refer the reviewer to Figure 8 in the appendix, where CoFie fails on super challenging thin structures, and we believe this is a common problem for any shape representation.
> > > >
> > > > 3DS2VS results (Fig. 8 in the paper) suggest that it is able to handle proposed shapes well if trained on sufficient amount of data.
> > > >
> > > > > Computational efficiency.
> > > >
> > > > I think number of local patches seen during the training is irrelevant in this discussion. My main concern that this model might be expensive to train on full ShapeNet or Objaverse. Rebuttal does not address it. That being said, this issue is way less concerning to me than weak evaluation so in the current state it does not affect my decision at all.

---

> > > > > ### Author Response · Authors · 2024-08-13
> > > > >
> > > > > We thank you for the detailed feedback. We will include more clarifications and experiment results to address your concerns.
> > > > >
> > > > > ---
> > > > >
> > > > > > Model capacity and training data size
> > > > >
> > > > > We would like to provide more context for the task of generalizable shape auto-decoding. In the table below, we compare the number of parameters for the representative generalizable shape auto-encoding (GAE) and shape auto-decoding (GAD) works.
> > > > >
> > > > > | Task | Model  | # Params in Encoder | # Params in Decoder |
> > > > > |------|--------|---------------------|---------------------|
> > > > > | GAE  | 3DS2VS | 4.2 M               | 101.9 M             |
> > > > > | GAD  | DeepLS | No Encoder          | 0.05 M             |
> > > > > | GAD  | CoFie  | No Encoder          | 0.06 M             |
> > > > >
> > > > > In the context of GAE, we agree that the encoder of 3DS2VS is not large. However, the situation becomes different compared with GAD methods.
> > > > >
> > > > >
> > > > > ---
> > > > >
> > > > > > NKSR and 3DS2VS training and evaluation
> > > > >
> > > > > We thank the reviewer for bringing the comment that using small data for training might not be sufficient for GAE methods, e.g. 3DS2VS. In general, we agree with the comment as these GAE methods learn priors of latent shape space under large data (acknowledged in our first rebuttal reply as “We agree that shape auto-encoding requires a large amount of training data as it learns priors to map the shape/point cloud into latent code. ”). In our submission, we train 3DS2VS with the 1000 shape for a fair comparison with GAD methods, and using 1000 shapes for training GAD methods are adopted in prior works, e.g. DeepLS.
> > > > >
> > > > > In our last rebuttal reply, we have included results of NKSR and 3DS2VS on the full set of 5 training categories in Table R1, where we use 20K shapes for training, which is comparable to original 3DS2VS paper. For a more comprehensive understanding, we include the training setting and performance on ShapeNet 10 **novel categories** as follows. Besides, the NKSR method is also trained with 20K shapes (see Table R2).
> > > > >
> > > > > |                 | 3DS2VS (our paper) | 3DS2VS (Table R1) | 3DS2VS (original paper) | CoFie (our paper) | CoFie (new) |
> > > > > |-----------------|--------------------|-------------------|-------------------------|-------------------|-------------|
> > > > > | **Task**        | GAE                | GAE               | GAE                     | GAD               | GAD         |
> > > > > | **#Training Shapes** | 1K            | 20K               | 55K                     | 1K                | 20K         |
> > > > > | **CD**          | 16.0               | 9.30              | /                       | 3.18              | 2.04        |
> > > > > | **gloU**        | 81.2               | 94.8              | /                       | 98.3              | 98.7        |
> > > > >
> > > > > The results implies that shape auto-encoding methods, e.g. 3DS2VS, suffer from limited generalization capability on novel categories, as the training distribution doesn't offer prior knowledge of them.
> > > > >
> > > > > Additionally, we note that we didn’t report performance of 3DS2VS (original paper) as its training categories overlap with our novel testing categories, which cannot be used for evaluating the generalization capability.
> > > > >
> > > > > ---
> > > > >
> > > > > > Performance of CoFie on ShapeNet, comparison with 3DS2VS/NKSR.
> > > > >
> > > > > We would like to point the reviewer to Table R1 in our last reply, where our method which trained with 1000 shapes has already demonstrated better performance than 3DS2VS/NKSR trained on 20K shapes. With 20K training shapes, CoFie demonstrates an even better performance.
> > > > >
> > > > > We additionally note that CoFie (generalizable shape auto-decoding) and 3DS2VS/NKSR (generalizable shape auto-encoding) are not directly comparable. The comparison above is in the context that the reviewer requested this comparison. We include the comparison between the two settings as follows for clarification.
> > > > >
> > > > > |                         | Input during inference                            | Latent Code                                 | Decoder            | Methods                     | Inference speed |
> > > > > |-------------------------|---------------------------------------------------|---------------------------------------------|--------------------|-----------------------------|-----------------|
> > > > > | **Generalizable Auto-Encoding** | On-surface points (w./w.o. additional point normals) | Predicted by the encoder in feed-forward | Large-size decoder | 3DS2VS, NKSR                | Seconds         |
> > > > > | **Generalizable Auto-Decoding** | SDF samples (not on-surface)                 | Optimized by back-propagation              | Small-size MLP     | DeepSDF, DeepLS, CoFie       | Minutes         |

---

> > > > > > ### Comment · Reviewer_V5fn · 2024-08-13
> > > > > >
> > > > > > Again, I really appreciate the commitment of the authors to the rebuttal and for providing additional details.
> > > > > >
> > > > > > > Number of parameters for compared models.
> > > > > >
> > > > > > I appreciate this clarification. This again confirms my concerns about evaluation and the choice of baselines. 3DS2VS has a very hight capacity decoder -- 200x larger than Cofie or DeepLS! To me, it means that might really struggle in low data regimes due to overfitting and thus training and evaluation on larger dataset is needed.
> > > > > >
> > > > > > > In our submission, we train 3DS2VS with the 1000 shape for a fair comparison with GAD methods, and using 1000 shapes for training GAD methods are adopted in prior works, e.g. DeepLS.
> > > > > >
> > > > > > See above. With such high capacity of the decoder, 3DS2VS can memorize all separate local geometries for 1000 training shapes hurting its ability to generalize. This model was never designed  for auto-decoding task or for low-data regime. Yet, it achieves very strong performance in auto-encoding task. I think that 3DS2VS is either not a good baseline for auto-decoding or all models should be evaluated in auto-encoding setting.
> > > > > >
> > > > > > > Additional results on 20K ShapeNet shapes in GAE and GAD settings.
> > > > > >
> > > > > > I appreciate the effort that went into these experiments. Unfortunately, I still think they are not convincing. As authors pointed out many times in previous replies GAE setting cannot be directly compared to GAD setting and I agree with this notion.
> > > > > >
> > > > > > > We additionally note that CoFie (generalizable shape auto-decoding) and 3DS2VS/NKSR (generalizable shape auto-encoding) are not directly comparable. The comparison above is in the context that the reviewer requested this comparison.
> > > > > >
> > > > > > I have not requested a comparison of GAE and GAD setting. My point was that 3DS2VS was designed as GAE model and its evaluation in GAD setting might not be fair comparison for 3DS2VS. My another point was that GAE models can very powerful if trained on sufficient data (GAE setting) and that maybe evaluation of the proposed method should be done in this setting instead of GAD.
> > > > > >
> > > > > > > The results implies that shape auto-encoding methods, e.g. 3DS2VS, suffer from limited generalization capability on novel categories, as the training distribution doesn't offer prior knowledge of them.
> > > > > >
> > > > > > In my experience, this statement might need additional empirical evidence. First, 3DS2VS architecture is based in on latent grid/array/cloud that learns local geometries which fosters generalization capabilities due to learning of local, simpler, geometric priors instead of global ones (Cofie utilizes the same powerful idea). Second, I have tested 3DS2VS and other auto-encoding methods based on latent grids in challenging out-of-distribution setting: models were trained on full ShapeNet and tested on large dataset of high-quality out-of-distribution meshes. 3DS2VS showed very strong results in this setting.

---

> > > ### Comment · Reviewer_V5fn · 2024-08-12
> > >
> > > I really appreciate the effort that authors are putting in the rebuttal. However, I still don't find proposed arguments convincing.
> > >
> > > > We note that the data size is closely related to the task and model capacity. The recent generalizable shape auto-encoding methods use large encoder networks, where larger data is necessary to train those large models.
> > >
> > > First, I want to point out that 3DS2VS (recent method) does not have 'large encoder network'. As encoder, it uses cross-attention with 1 layer (!) and 1 head (!) ([see source code, LL 204-207](https://github.com/1zb/3DShape2VecSet/blob/master/models_ae.py)). The majority of capacity of this model actually concentrated in the decoder.  That being said, I agree that these methods usually perform better if trained on large amount of data. This brings us to my second point: strong performance on small data is not indicative about strong performance on large scale data. Large capacity model can perform worse on small scale data due to overfitting but strongly outperform low capacity model on large scale data. To me, strong results on 1000 shapes are not indicative of potentially strong performance on large scale data.
> > >
> > > Also, paper does not include direct comparison to capacity of the proposed models (in terms of mln parameters), so I am not even sure that Cofie has less parameters than baselines.
> > >
> > > > We perform experiments on ShapeNet-15 categories (5 categories for training, and 15 categories for testing); comparison to NKSR
> > >
> > > Again, as I mentioned above, my concern is that strong performance of the proposed model on small scale training might not be indicative of performance on large scale data. In the end, one of the common application of the proposed method would be either large scale reconstruction or using this representation as input to generative models. Both of these applications require large-scale training and proposed results don't convince me as a person who trained reconstruction and generative models on full ShapeNet.
> > >
> > > > Additional evalution and metrics on ShapeNet.
> > >
> > > First, I appreciate authors adding additional metrics. I strongly recommend including them in the revised version of the paper. Second, I want to point out that these metrics are important: as your results suggest, good CD performance does not mean god IoU performance and vice versa.  Finally, I want to point out that these results are still not looking convincing to me because training was done on 1000 shapes. What will happen if we will retrain proposed method and other baselines on 1000 other shapes? Will results be the same? I think that even if we talk about evaluation on 1000 shapes, the training should be done across 5 different subsamples of the data, results averaged and variance reported along with mean (similar to what people do with few-shot learning models). Since model takes 1 day to train, this evaluation is absolutely feasible given authors computational resources and can serve a strong argument in favor of the proposed method.
> > >
> > > > Additional metrics on Thingi10K;
> > >
> > > Again, I appreciate the effort that went into these results. Unfortunately, they validate my concerns about weak evaluation of the paper. This additional evaluation shows that Cofie might not be as strong as was initially presented: both NGLOD and UODFs have better performance both in teroms of gIOU and CD after additional evaluation. My concern is that similar pattern will hold for ShapeNet results if trained with stronger baselines and larger data.
> > >
> > > SIDENOTE. After discussion, I think that maybe this representation can be useful for text-to-3D or image-to-3D distillation but this is not evaluated as well. This does not affect my rating -- just a note that I thought might be useful to authors.

---

> ### Author Response · Authors · 2024-08-13
>
> > Performance of CoFie on Thingi, comparison with NGLOD/UODFs
>
> We would like to clarify NGLOD/UODFs are shape-specific auto-decoding methods. Different from generalizable shape auto-decoding where only the latent shape representation is optimized, NGLOD/UODFs also optimize the model for each specific shape. Thus, shape-specific auto-decoding methods naturally have better performance. We include the comparison of the two settings as follows.
>
> |                             | Latent representation         | Decoder                | Methods                        | Inference speed |
> |-----------------------------|-------------------------------|------------------------|--------------------------------|-----------------|
> | **Generalizable Auto-Decoding** | Optimized during inference  | Fixed weight after training  | DeepSDF, DeepLS, CoFie         | Minutes         |
> | **Shape-specific Auto-decoding** | Optimized during inference | Optimized for each shape | NGLOD, UODFs                   | Hours           |
>
> The close performance between CoFie and NGLOD/UODFs in Table 2R demonstrates the effectiveness of our work. We also report the performance of CoFie trained by 20K shapes as follows based on Table R2, where CoFie is further improved with more training data.
>
> |                            | CoFie (our paper) | CoFie (new) | NGLOD                 | UODFs                 |
> |----------------------------|-------------------|-------------|-----------------------|-----------------------|
> | **Task**                   | GAD               | GAD         | SSAD                  | SSAD                  |
> | **#Training Shapes**       | 1K                | 20K         | Each testing shape    | Each testing shape    |
> | **CD**                     | 1.87              | 1.22        | 1.04                  | 0.932                 |
> | **gloU**                   | 99.0              | 99.2        | 99.3                  | 99.4                  |
> | **Inference Time/Shape**   | 10 min            | 10 min      | 105 min               | 300 min               |
>
> Similarly, we would like to clarify the two kinds of methods (GAD v.s. SSAD) cannot be directly compared. This analysis is performed as it was required by the reviewer for understanding CoFie’s performance.
>
> ---
>
> We thank the reviewer for the suggestions on understanding the performance of CoFie in the context of recent generalizable shape auto-encoding and shape-specific auto-decoding works. Although we don't agree the methods of different settings are directly comparable, we hope the analysis could provide better understanding of our model. We also hope the added results and analysis could clarify the effectiveness of CoFie in the context of using more data for training. We appreciate the time and effort spent in the discussion period, and we will incorporate the discussion content and all your suggestions in the revision.

---

> > ### Comment · Reviewer_V5fn · 2024-08-13
> >
> > Again, I appreciate the effort that went into additional results.
> >
> > > Similarly, we would like to clarify the two kinds of methods (GAD v.s. SSAD) cannot be directly compared.
> >
> > I agree and I never said that the methods should be directly compared. Honestly speaking, I think majority of the discussion we are having right now stems from the fact that initial submission mixes different models in the first place. For example, Tables 1 and 3 has 3DS2VS (GAE model); DeepSDF, DeepLS and Cofie (GAD models). In my opinion, it is not correct comparison on the first place because 3DS2VS was designed as GAE model and authors themselves highlighted many times that GAE and GAD models are not directly comparable.
> >
> > Similarly, Table 3 has NGLOD (SSAD model); DeepSDF (or DeepLS judging by update) and Cofie -- both GAD models. Again, authors themselves stress out that these models are not directly comparable (and I agree with that!), yet this comparison is in the current version of the paper.
> >
> > In conclusion, I really like the proposed method and I think it might be very powerful if trained on sufficient data in auto-encoding setting. But evaluation in the current iteration of the method feels all over the place: GAE and GAD models are mixed for evaluation; GAD an SSAD models are mixed as well. Personally, I am not even sure that GAD setting is the proper way to evaluate shape representations (for me, GAE setting offers more practical advantages) but this does not hold a weight in my decision.
> >
> > I also want to stress out again that I really appreciate the effort the authors put in this discussion.

---

> > > ### Author Response · Authors · 2024-08-13
> > >
> > > Thanks for the reply. We would like to conclude the latest discussion as follows.
> > >
> > > - We are happy the reviewer finally agree with us that GAE, GAD, SSAD methods are not directly comparable.
> > > - For the GAE method 3DS2VS, training it on 1000 shapes in the original paper is not convincing enough. The reviewer think the training set is too small and suggested us to train with larger data. We agreed with this point and we provided new results.
> > >
> > > With the updated content, what we achieved is
> > > - Best GAD performance, no matter in low-data (1000 training shape) or larger-data (20K, added during rebuttal) regime.
> > > - Similar performance with SoTA GAE and SSAD methods (3DS2VS/NKSR trained with 20K shapes, NGLOD/UODFs). I would say the performance is "similar" rather than "comparable", as they are not directly comparable.
> > >
> > > ---
> > >
> > > Regarding your question why we put NGLOD, DeepLS, CoFie together in tables, we agree this might lead to confusion. However, we have noted in the paper Line 243 for categorizing generalizable and shape-specific models. In Line 247 we clarify the setting of generalizable shape auto-encoding and noted it is not directly comparable. We also note classify each method, as in Line 251, 253, 255, 258. But we appreciate the feedback from the reviewer, and we will also group the methods in the tables for better clarity.
> > >
> > > ---
> > >
> > > Finally, the reviewer suggested
> > > > I really like the proposed method and I think it might be very powerful if trained on sufficient data in auto-encoding setting.
> > >
> > > We would like to clarify that this method is designed for GAD. Thus, it is not applicable to GAE, as the coordinate field need to be updated by optimization (equation 8 and 9). We are not able to supervise the prediction of coordinate field if we use a feed-forward model to do GAE, as we don't have the ground truth for the coordinate field. GAE, GAD and SSAD are different tasks, **requiring the model designed for one task can be applied to other tasks is not rational**. Research on how to build an unified framework for GAE/GAD/SSAD is promising, but this is out of our current scope.
> > >
> > > ---
> > >
> > > For your comment
> > > > Personally, I am not even sure that GAD setting is the proper way to evaluate shape representations (for me, GAE setting offers more practical advantages)
> > >
> > > We agree GAE has the advantage of i) can learn general priors with large model and large data, ii) the learned latent space can be directly used for downstream tasks, e.g. generation. But we would like to clarify that every task has its own value. The GAD task benefits from its compactness, lower requirement on training data and model size.
> > >
> > > We appreciate the detailed feedbacks and the insights on GAE method from the reviewer.

---

> > > > ### Comment · Reviewer_V5fn · 2024-08-13
> > > > **final reply to authors**
> > > >
> > > > I think we have reached limits of constructive discussion and further engagement will be waste of everyone's time.
> > > >
> > > > I am not going to change my rating. I strongly believe that evaluation of the paper is flawed and if we accept it in the current state, we are going to normalize this evaluation and thus will create a bad precedent for follow-up work. Authors have failed to convince me that it is not the case. Below is summary of my concerns based on my initial review and discussion.
> > > >
> > > > **1. Tables 1-2 provide comparison with either very old models (DeepSDF, 2019; DeepLS, 2020) or with models that were not designed for GAD setting(3DS2VS)**. As I pointed out in my initial review and many times during the discussion, 3DS2VS was designed for GAE setting and authors acknowledged this in the discussion. I think that choice of baselines in this experiment is weak and favors proposed method. I think that evaluation should be done in GAE setting (preferably on full ShapeNet) because GAD setting is rarely used these days and majority of novel shape representation methods evaluate in GAE setting.
> > > >
> > > > **2. Even if we consider GAD setting, models are evaluated based on training on 1000 shapes.** This is a low data regime with potentially high variance in evaluation results if training and evaluation is done using different samples of 1000 shapes. I don't think that this evaluation is enough to support the claim that proposed method generalizes better than (weak) baselines. Even if paper uses low-data regime setting for evaluation, results should be presented for at least 5 different runs on different subsamples of the ShapeNet.
> > > >
> > > > **3. Choice of baselines in Table 3 has the same issue as choice of baselines in Tables 1-2**. NGLOD is SSAD model, DeepLS and Cofie are GAD models. Again, this is not a correct comparison and thus is not enough to support claims in the paper. I think that either proposed method should be evaluated in SSAD setting, or this evaluation should be removed.
> > > >
> > > > I am not going to reply to further comments by authors. I think that this thread has provided enough feedback to both authors to revise their submission and other reviewers/ACs to help reach final decisions.

---

> ### Author Response · Authors · 2024-08-13
>
> Thanks for the feedback. We would like to conclude the discuss as follows.
>
> > Tables 1-2 provide comparison with either very old models (DeepSDF, 2019; DeepLS, 2020) or with models that were not designed for GAD setting(3DS2VS).
>
> - We agree that DeepSDF and DeepLS are old methods. In the meantime, we kindly remind that **they are the only two directly comparable methods for GAD** as far as we know. We are happy to compare with any baseline for GAD if you could provide any references. Whether DeepSDF and DeepLS are old is orthogonal to our contributions in this work, as our contributions are ablated.
> - **We note we have mentioned many times GAE and SSAD methods are used as reference and they are not directly comparable methods.** See Line 243 for categorizing generalizable and shape-specific models. In Line 247 we clarify the setting of generalizable shape auto-encoding and noted it is not directly comparable. We also note classify each method, as in Line 251, 253, 255, 258.
> - **The reviewer got an wrong impression by him/herself and urged us comparing with GAE methods** because him/herself works on GAE (mentioned as "as a person who trained reconstruction and generative models on full ShapeNet", where "reconstruction" is GAE)  and this leads to the biased reviews towards GAE. **Again, we never claim we should compare GAD methods to GAE and SSAD methods. Instead, this is the understanding of the reviewer.** Please see the initial review of V5fn for the details of requesting GAE method experiments.
>
> > Even if we consider GAD setting, models are evaluated based on training on 1000 shapes.
> - We acknowledge that we trained the model with 1000 shapes.
> - **Using 1000 shapes is standard in prior works.** See DeepLS.
> - At the same time, we would like to point the review to the results provided during the rebuttal period of training CoFie on 20K shapes.
> - **The new experiment demonstrated that CoFie shows good performance in the context for both low-data and larger-data regime.**
>
> > Choice of baselines in Table 3 has the same issue as choice of baselines in Tables 1-2.
> - **This is your own fault for comparing these methods together.** Again, we note **we have mentioned many times** GAE and SSAD methods are used as reference and they are not directly comparable methods. See ine 243 for categorizing generalizable and shape-specific models. In Line 247 we clarify the setting of generalizable shape auto-encoding and noted it is not directly comparable. We also note classify each method, as in Line 251, 253, 255, 258.
> - The reviewer didn't realize the differences between GAE, GAD, SSAD until the last reply. Even after he/she realized this, he/she still attacked us using his/her own wrong understanding.
> - Again, at the very beginning, we note we have mentioned many times GAE and SSAD methods are used as reference and they are not directly comparable methods. See ine 243 for categorizing generalizable and shape-specific models. In Line 247 we clarify the setting of generalizable shape auto-encoding and noted it is not directly comparable. We also note classify each method, as in Line 251, 253, 255, 258.

---

### Official Review · Reviewer_LpYZ · 2024-07-15

**Soundness:** 2
**Presentation:** 2
**Contribution:** 3
**Rating:** 6
**Confidence:** 4

**Summary:**

The paper proposes a local prior based method where the model is trained on a dataset and learns a prior over local patches of the shapes, and then at test time reconstructs patches by optimising to minimise the reconstruction error w.r.t. the trained model's prior. While this has already been done in the literature, they propose to learn a coordinate transformation for each local region and initialise the coordinate transformation based on the geometry in that region. They also use a quadratic layer at the end of the network, inspired by their analysis of representing local patches by a quadratic surface.

**Strengths:**

- Good ablation study showing the benefits of each component. Interestingly the geometric initialization is very important for performance improvement, suggesting that the latent based model has trouble training efficiently.

**Weaknesses:**

- A lot of things are not made clear (asked in the questions) which makes it hard to understand the method
- As mentioned in the strengths, it seems that the benefit of this method is that the coordinate field (and especially its init) it enables the latent based model to train better, rather than being a better representation. This could be explored more deeply. Also it makes the question below ("It seems the proposed method has no regularization, is there a reason for this?")

**Questions:**

- The method is called coordinate field, but it is not actually a true coordinate field: the coordinate transformation is learned discretely as parameters for each voxel, not continuously throughout the domain?
- Usually shape autodecoding uses regularization on the latent parameters so then when doing test time latent optimisation the latent parameters can be initialized to zero and be "close" to many possible shapes. It seems the proposed method has no regularization, is there a reason for this?
- Line 219: "compute the derivatives of the SDF" however at test time you don't have access to the full SDF (or do you?), so how to you get well oriented normals for initialization?
- Would like more information about what SDF supervision exactly is given at test time. Is it just the surface points, or is assumed the SDF is known at test time as well?
- Would like timing performance of test time optimization (general ballpark and if this compares to other methods)
- Line 243: two types of methods or three types of methods?
- Missing ref on line 114, line 128 and line 197

**Limitations:**

- Yes, limitations have been discussed adequately.

---

> ### Author Rebuttal · Authors · 2024-08-07
>
> We thank you for the positive comments and your acknowledgment of the thoroughness of our ablation study. We will address your comments as follows.
>
> ---
>
> **[W2, Benefit of Coordinate Field]** We kindly note that the coordinate field is a part of our representation, as the coordinate field is involved in the computation of decoding shape surfaces. It makes the learning more efficient, improving the geometry modeling capability, and helping us learn the MLP in a more compressed local patch geometry space.
>
> ---
>
> **[Q1, Coordinate field]** A field is defined as follows in Wikipedia: “In science, a field is a physical quantity, represented by a scalar, vector, or tensor, that has a value for each point in space and time.” Our coordinate field is a discrete field, where the coordinate frame value of a 3D point is associated with the local patch it belongs to. We note that this discretization is the key to defining local patches and enabling the geometry-aware coordinate field and its initialization.
>
> ---
>
> **[Q2, Regularization]** Thanks for pointing this out. We use the same regularization as previous works that penalize the latent code values, following the standard regularization term in DeepSDF and DeepLS. We didn’t include the term in the loss function as it is standard. We will revise the text for clarity.
>
> ---
>
> **[Q3, Well-oriented normal]** Great question! We clarify that at test time, we don’t have full SDF but SDF point samples (following DeepSDF, DeepLS, NGLOD, etc), and the estimated normal can be imperfect. Thus, as we mentioned in equation 9, we optimize the latent code and the coordinate frame jointly, which makes our model more robust to noise in the estimated normals.
>
> ---
>
> **[Q4, Supervision at test time]** During testing, we follow practices from DeepSDF, DeepLS, and NGLOD. We sample points (with known SDF values), including points close to the surface, which is a standard point sampling method for shape auto-decoding. Thus, the points are not on-surface points, but also we don’t have access to the full SDF (we only have point samples of the SDF). In evaluation, we compare with DeepSDF, DeepLS, and NGLOD inference on the same data with our method.
>
> ---
>
> **[Q5, Inference time]** We achieve an inference time of 10 minutes in the setting of *optimizing a single shape*, which is much faster than the baseline NGLOD (105 min). Moreover, our method benefits from the capability of *inference on a batch of shapes* by sampling batched local shapes from all of them. This property further improves the efficiency of our method. NGLOD does not support batched inference since it optimizes a non-regular grid which is different for each test shape.
>
> ---
>
> **[Q6, L243]** Thank you for catching that! We meant three types of methods: 1) generalizable shape auto-decoding methods, 2) shape-specific shape auto-decoding methods, and 3) generalizable shape auto-encoding methods, which are used as a reference and are not directly comparable (as discussed in L247). We will clarify this in the revision.
>
> ---
>
> **[Q7, missing references]** Thanks for the detailed feedback. These references include previously mentioned works including [22,38,8,34,30]. We will revise the references in revision.

---

> > ### Comment · Reviewer_LpYZ · 2024-08-11
> >
> > Thank you for your replies, they have addressed my concerns (and I hope they are clarified in the revised paper). I have also read the concerns of other reviewers and your replies, and am happy with them. I maintain my previous position of weak accept.

---

> > > ### Author Response · Authors · 2024-08-11
> > >
> > > We are glad to see that the reviewer's concerns were resolved, and we will update our paper based on our discussion with the reviewer. Thank you!

---

### Author Rebuttal · Authors · 2024-08-07

We would like to thank the reviewers for their detailed feedback. We received positive comments from the reviewers who appreciated our clear ablation and convincing evaluation (LpYZ, Kq6m), the efficiency of our method (V5fn, 4hsy), good performance (V5fn, 4hsy), clear theoretical proof (4hsy, Kq6m), novelty (Kq6m). Our goal is to **introduce geometric insights when developing 3D neural representations**, which have proven to be effective in the experiments presented in the paper. Moreover, this paper follows a theoretical style, in which we provide rigorous insights on how to develop 3D representations.

The reviewers also had some additional questions and comments that we address independently in the answer for each reviewer. Moreover, we include a one-page pdf with some further explanation of equation 14 in our proof. Alternatively, **we noticed some of the reviewers misunderstood our setting**, where we are doing shape auto-decoding following DeepSDF and DeepLS, instead of shape auto-encoding, as mentioned in related works like 3DS2VS, NKSR, etc. Please see the details in our rebuttal.

---

### Author Response · Authors · 2024-08-14
**Additional Comments**

Dear All,

             We think the reviewers are split into two groups. We do understand the reasons. On one hand, 3D neural representations and 3D generation are quite crowded areas now. To publish a paper, one has to have amazing results and compare to many baselines. This was the reason of two negative reviews. We do admit that we did not go to the level of typical papers in this area now in terms of baseline comparisons.

            However, we believe that our paper opens a new direction in 3D representations which seek to develop geometry-aware representations. All existing work in this space follow 2D strategies under some 3D representations. These approaches, in our view, are black box. We have little understanding on what are learned. The only way to convince the reviewers is to show amazing figures and extensive comparisons, many of those have overfitting behaviors. From a scientific point of view, we do not think this is necessarily good thing. A paper should not be evaluated just on experimental results. Our approach is based on  simple and effective geometric insights, which are based on rigorous analysis. We hope it will simulate future research in this direction. Most of questions raised by the reviewers can be done as follow-up work either by ourself or others.

           If this paper was done 2-3 years ago, then we would not have these evaluation issues. However, we should not be penalized because of this, as it opens a new direction in 3D representations.

---

### Decision · Program_Chairs · 2024-09-25

**Decision:**

Accept (poster)

**Comment:**

The paper received a very large number of posts between authors and reviewers after the rebuttal, and also between reviewers during the post-rebuttal discussion. All reviewers agreed on the technical and intellectual novelty of the paper. The AC also read the paper and appreciated the geometric insight of attaching learnable local coordinate frames to each patch and approximating the surface through a composition of local, frame-aligned MLPs with quadratic layers. Two of the reviewers remained negative because of limitations in the evaluation i.e., the evaluation should be better done in the generalized auto-encoding setting and preferably on full ShapeNet. The AC agrees that the evaluation could have been more bulletproof, yet, the novelty and geometric insights deserve recognition. The AC recommends acceptance, yet urges the authors to incorporate their revised comparisons and additional experiments in larger datasets in the final version of the paper! The authors should also acknowledge the limitations of their evaluation, including the limitation of using shallow MLPs with ReLU activations in the comparisons (instead of more modern variants, such as GLUs/GEGLUs). The paper should also correct issues with citations, and also add citations to more recent works on 3D neural generative geometric representations e.g., CLAY, Make-A-Shape, GEM3D, Mosaic-SDF.